# Electronic origin of reorganization energy in interfacial electron transfer

Sonal Maroo[1], Leonardo Coello Escalante[1], Yizhe Wang[1], Matthew P. Erodici[1], Jonathon N. Nessralla[1], Ayana Tabo[1], Takashi Taniguchi[2], Kenji Watanabe[3], Ke Xu[1], David T. Limmer[1,4,5,6 ✉] & D. Kwabena Bediako[1,4,6 ✉]

Electron transfer (ET) reactions underpin energy conversion and chemical transformations in both biological[1,2] and abiological[3–5] systems. The efficiency of any ET process relies on achieving a desired ET rate within an optimal driving force range. Marcus theory[6,7] provides a microscopic framework for understanding the activation free energy—and therefore the rate—of ET in terms of a key parameter: the reorganization energy. For electrified solid–liquid interfaces, it has long been conventionally understood that only factors in the electrolyte phase are responsible for determining the reorganization energy and that the electronic density of states (DOS) of the electrode only serves to dictate the number of thermally accessible channels for ET[5,8–12]. Here we show instead that the electrode DOS plays a central role in governing the reorganization energy, far outweighing its conventionally assumed role. Using atomically layered heterostructures, we tune the DOS of graphene and measure outer-sphere ET kinetics. We find the ensuing variation in ET rate arises from strong modulation in a reorganization energy associated with image potential localization in the electrode. Here we redefine the traditional paradigm of heterogeneous ET kinetics, revealing a deeper role of the electrode electronic structure in interfacial reactivity.

In the initial formulation of Marcus theory for homogeneous electron transfer (ET) involving redox-active ions in solution, the activation free energy was explained in terms of a reorganization energy ($\lambda$) penalty required to distort the atomic configuration and solvation environment of the reactant species to resemble those of the product state[6,7]. Extensions in the so-called Marcus–Gerischer[8] and Marcus–Hush[5,9] formalisms rationalized heterogeneous ET processes at electrode–electrolyte interfaces, specifically addressing the ET rate constant in the weak coupling limit. The quantum mechanical theory of ET—pioneered by Levich, Dogonadze, Chernenko and Kuznetsov[11,12]—similarly leads to Marcus–Hush-type rate expressions. The seminal adaptation of the Marcus–Hush model by Chidsey[10] explained the dependence of interfacial ET rates on driving force and temperature by incorporating the Fermi–Dirac distribution of occupied electronic states in the electrode. In all of these and later[13,14] extensions that incorporated a non-uniform (energy-dependent) density of electronic states (DOS) profile of the electrode, the DOS of the electrode exclusively serves to dictate the number of thermally accessible channels for ET. Furthermore, consistent with the original framework, $\lambda$ is presumed to arise largely from nuclear reconfigurations in the electrolyte phase (typically those of the solvent and in some cases the redox molecule itself).

Enhancements in interfacial charge transfer at electrodes and photoelectrodes due to electrostatic variations in carrier doping or defects (including vacancies, step edges and grain boundaries) are commonly explained as arising from increases in the electrode DOS near the Fermi level ($E_F$), ostensibly due to increased number of thermally accessible channels for ET[15–17]. Yet the Marcus–Hush–Chidsey (MHC) or Marcus–Gerischer framework often fails to quantitatively predict interfacial ET rate constants, even for relatively simple electrode reactions—overestimating ET rates by an order of magnitude or more[18–20]. A recent example of these discrepancies is found in the interfacial ET behaviour of twisted-bilayer graphene and twisted-trilayer graphene[14,21]. In these moiré electrode systems, which display periodic spatial localization of the electronic charge density in moiré superlattice topological defects, models derived from the MHC framework are unable to account for the large variation in ET rate with twist angle despite modifications to account for a DOS profile that varies with energy[13] as well as quantum capacitance effects that lead to changes in DOS of the electrode following electrochemical polarization[14,21].

What is missing from these frameworks to explain how electrode defects produce such strong local enhancements in interfacial ET is a consideration of the electrode's contribution to $\lambda$. Yet, molecular dynamics simulations have long predicted that $\lambda$ for interfacial ET would vary with the distance of a redox molecular ion from a metallic electrode, owing to image charge interactions[19,22,23]. First-principles calculations of ET rates at graphite electrodes also considered a contribution of the electrode dielectric response to $\lambda$, but the effect was presumed to be sufficiently small to be neglected[24]. Moreover, recent simulations

[1]Department of Chemistry, University of California, Berkeley, Berkeley, CA, USA. [2]Research Center for Materials Nanoarchitectonics, National Institute for Materials Science, Tsukuba, Japan. [3]Research Center for Electronic and Optical Materials, National Institute for Materials Science, Tsukuba, Japan. [4]Chemical Sciences Division, Lawrence Berkeley National Laboratory, Berkeley, CA, USA. [5]Materials Sciences Division, Lawrence Berkeley National Laboratory, Berkeley, CA, USA. [6]Kavli Energy NanoScience Institute, Berkeley, CA, USA. ✉e-mail: dlimmer@berkeley.edu; bediako@berkeley.edu

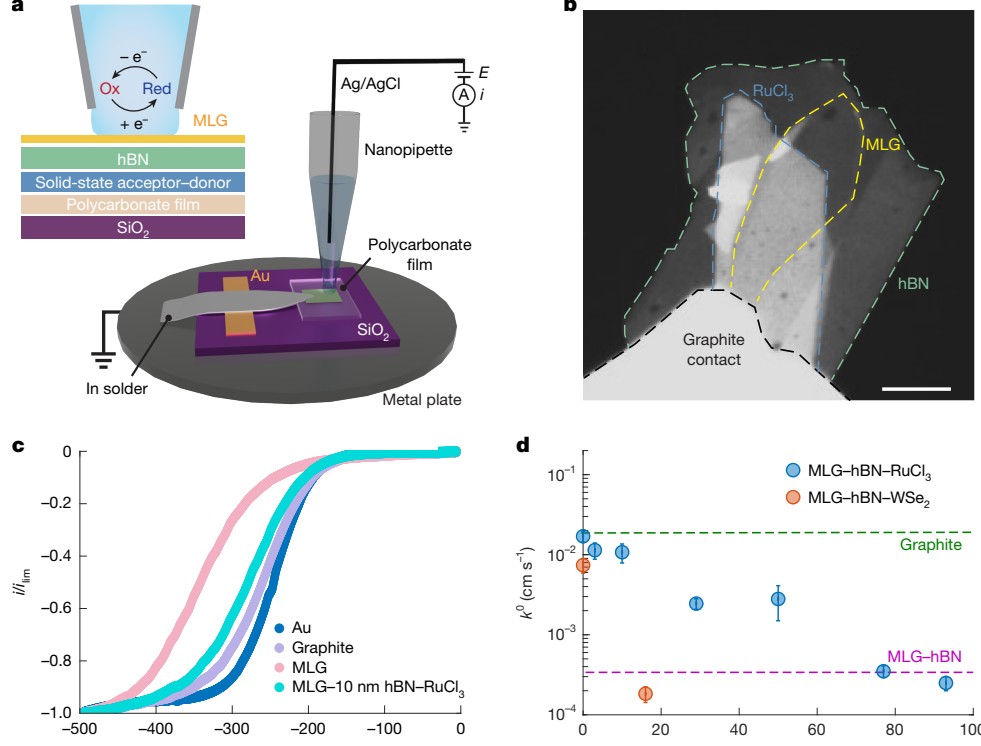

**Fig. 1 | Electrochemistry of MLG–hBN–crystalline donor–acceptor heterostructures. a**, Schematic illustrations of electrochemical measurement at MLG surfaces using SECCM. **b**, Optical micrograph of a device fabricated from an exfoliated MLG flake on hBN and RuCl$_3$. **c**, Representative steady-state voltammograms of 2 mM Ru(NH$_3$)$_6^{3+}$, depicting the mean current from the forward and backward sweeps, in 0.1 M KCl solution obtained at gold, graphite and MLG–hBN heterostructures (with and without α-RuCl$_3$). Scan rate, 100 mV s$^{-1}$.

The current and limiting current are denoted by $i$ and $i_{lim}$, respectively. **d**, Dependence of the interfacial ET rate constant, $k^0$, on the thickness of hBN spacer between MLG and RuCl$_3$ (crystalline acceptor) or WSe$_2$ (crystalline donor). Each data point represents the mean of multiple measurements for samples with a given hBN thickness; error bars indicate the s.d. for each $k^0$, where $n$ varies from 3 to 6. Scale bar, 10 μm (**b**).

of twisted-bilayer graphene have established a connection between the moiré twist angle and the screening of charge carriers within the electrode, identifying that a twist-angle-dependent reorganization energy can account for the interfacial ET behaviour observed at moiré graphene electrodes[25,26]. A unique aspect of these electrode systems is the ability to continuously change the density of states (DOS) at the Fermi energy, and correspondingly tune the ability of the electrode to screen electric fields. The Thomas–Fermi (TF) screening length ($\ell_{TF}$) quantifies the length-scale over which charges are screened in imperfect metals. As $\ell_{TF}$ scales inversely with DOS, higher metallicity leads to sharper charge localization, whereas lower metallicity yields a more diffuse charge distribution. Such tunability offers a new avenue to investigate how electronic screening shapes interfacial ET[27].

Here we directly and systematically probe the DOS dependence of interfacial ET using van der Waals assembly of atomically thin crystals. Using solid-state dopant layers and hexagonal boron nitride (hBN) spacers, we electrostatically tune the doping levels in monolayer graphene (MLG) and measure the resulting variation in heterogeneous outer-sphere electrochemical ET rates of the [Ru(NH$_3$)$_6$]$^{3+/2+}$ couple. The strong variation in graphene charge density with changes to hBN spacer thickness is shown to be mediated by defects in the hBN crystals. A continuum model is leveraged to obtain a microscopic understanding of the dependence of interfacial ET rate on the electrode DOS. We find that the ensuing variation in ET rates with charge carrier density cannot be modelled in the Marcus framework when one only considers the change in the number of thermally accessible channels for ET. Instead, our measurements and simulations unveil the considerably more dominant DOS-dependent reorganization energy, which accurately captures the large experimental variation in interfacial ET rate

with DOS. We observe that at low charge carrier densities—such as those found in many low-dimensional electrode materials, as well as bulk or nanocrystalline semiconductors—the reorganization energy penalty owing to the low electrode DOS can be of a magnitude comparable with that arising in the solvent at a metallic electrode. This systematic study of the DOS dependence of interfacial ET rates on well-defined electrode surfaces challenges the conventional paradigm that reorganization energy contributions predominantly arise from the electrolyte side of the electrode–electrolyte interface and establishes a general microscopic framework for understanding heterogeneous ET that explicitly accounts for the electronic properties of the electrode in governing the free energy of activation.

## Measurements of interfacial ET as a function of DOS

Isolating two-dimensional (2D) crystals enables the assembly of van der Waals heterostructures with tailored electronic and chemical properties[28,29]. When 2D crystals with disparate work functions are interfaced, the resulting electric field leads to charge transfer and interfacial doping, analogous to the effect of an electrostatic gate in a field effect transistor; past work has shown that α-RuCl$_3$ and WSe$_2$ can thus be used to dope graphene with holes and electrons, respectively[30]. This approach provides a modular doping mechanism similar to vacancy or substitutional doping but avoids introducing chemical disorder into the active layer. Such heterostructures provide an exceptionally well-defined platform for examining how doping-induced DOS changes impact the rate of ET.

We fabricated mesoscopic electrochemical devices comprising MLG on RuCl$_3$. We also fabricated samples with hBN spacers placed between MLG and RuCl$_3$, and varied the hBN spacer thickness from 3 nm to 120 nm

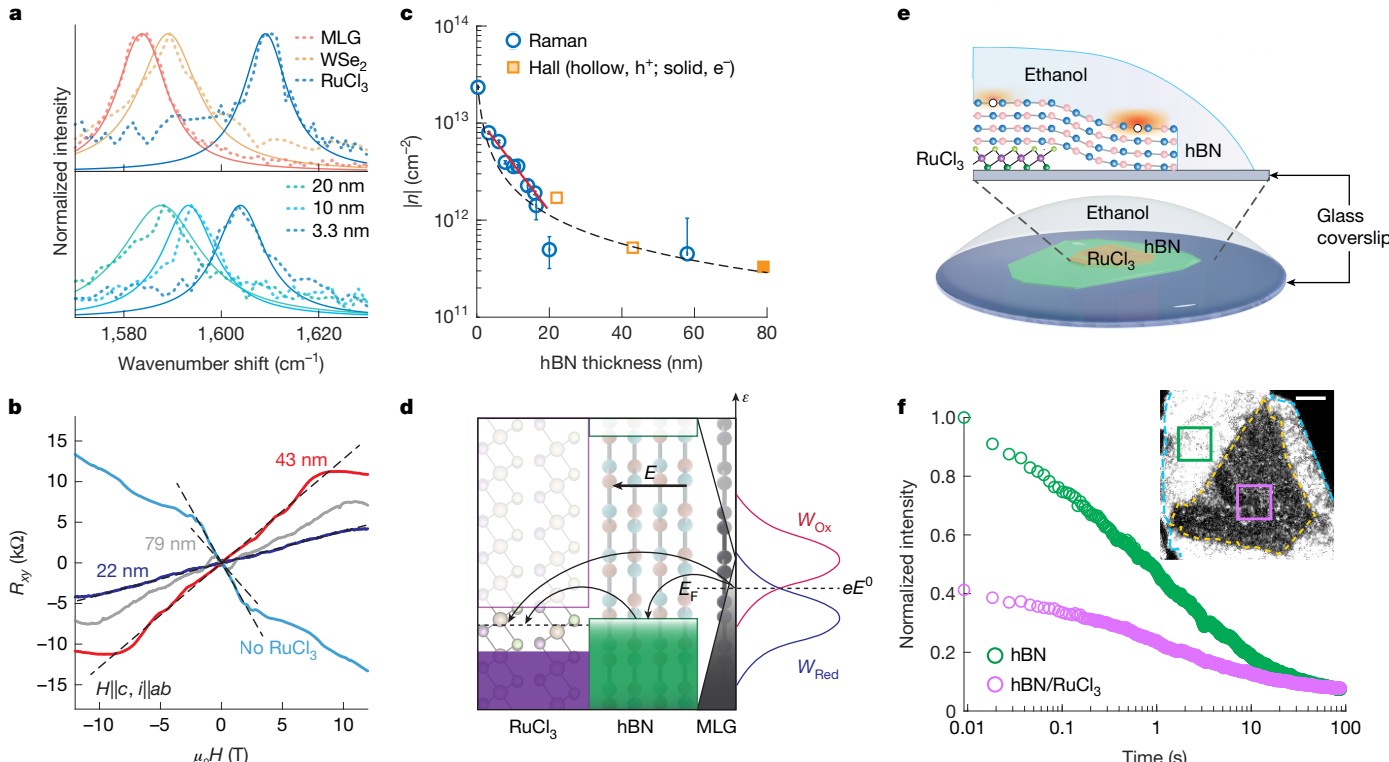

**Fig. 2 | RuCl$_3$-induced doping in MLG and quenching of hBN fluorescence.**
**a**, Top: Raman G-peak spectra of MLG−hBN, MLG−WSe$_2$ and MLG−α-RuCl$_3$
heterostructures. Bottom: G-peak spectra of MLG−hBN−α-RuCl$_3$
heterostructures with varying hBN thicknesses. Solid lines indicate Voigt fits
from which peak positions are obtained. **b**, Hall resistance, $R_{xy}$, as a function
of magnetic field at 1.8 K for three hBN thicknesses in MLG−hBN−α-RuCl$_3$
heterostructure devices, compared with undoped MLG. $\mu_0H$ represents the
magnetic flux density, where $\mu_0$ is the vacuum permeability and $H$ is the
magnetic field strength. '$H\|c$, $i\|ab$' indicates that $H$ is aligned parallel to
the crystallographic $c$ axis of graphene, and that current $i$ is flowing in the
$a$–$b$ plane. **c**, Absolute carrier density in MLG, $|n|$, as a function of hBN spacer
thickness in MLG−α-RuCl$_3$, derived from Raman G-peak shifts and Hall
measurements, compared with $|n|$-predicted first-principles calculations
(dashed black line[32]). A polynomial fit (solid red line) phenomenologically

models the sub-20 nm regime in which enhanced doping deviates from classical
screening, due to defect-mediated charge transfer. Error bars indicate the s.d.
for each $|n|$, where the number of data points for each $|n|$ varies from 6 to 10.
**d**, Schematic illustration of band alignment and interfacial charge transfer
between graphene and α-RuCl$_3$, depicting $E_F$ shifts relative to its band structure
and corresponding DOS modifications. $W_{Redox}$ denotes redox molecule
probability distributions ($W_{Ox}$, oxidized; $W_{Red}$, reduced). $e$ denotes the elementary
charge and $E^0$ denotes the standard reduction potential of the redox couple.
**e**, Illustration of the experimental set-up for liquid-induced fluorescence
measurements in hBN−α-RuCl$_3$ heterostructures. **f**, Normalized hBN emission
intensity versus illumination time in regions with and without α-RuCl$_3$.
The inset displays a wide-field fluorescence image (561 nm laser, ~5 kW cm$^{-2}$,
6 ms exposure). Scale bar, 5 µm.

(Methods and Extended Data Fig. 1 for further details on sample fabrica-
tion and characterization). Electrochemical measurements were con-
ducted using scanning electrochemical cell microscopy (SECCM), which
enables nanoscale electrochemical measurements by positioning an
electrolyte-filled nanopipette over the sample and forming a confined
electrochemical cell following meniscus contact[31]. Here we used quartz
nanopipettes approximately 600–800 nm in diameter (Extended Data
Figs. 2 and 3) containing 2 mM hexaammineruthenium(III) chloride and
100 mM potassium chloride as supporting electrolytes.

Figure 1a,b presents schematics of the sample and measurement
set-up alongside an optical micrograph of a representative MLG−
hBN−RuCl$_3$ device in contact with a graphite flake, which serves as
the electrical contact. Steady-state cyclic voltammograms of the
[Ru(NH$_3$)$_6$]$^{3+/2+}$ couple at the basal plane of a MLG−10-nm-hBN−RuCl$_3$
heterostructures (Fig. 1c) reveal a shift in the half-wave potential $E_{1/2}$
to more positive potentials compared with cyclic voltammograms
of analogous samples without RuCl$_3$, consistent with enhanced DOS
from RuCl$_3$-induced hole doping that facilitates the electroreduction
of [Ru(NH$_3$)$_6$]$^{3+}$. Notably, even with a 10 nm (25-layer) hBN spacer,
MLG displays ET kinetics approaching those of graphite and has
nearly reversible electrochemical behaviour at the basal plane. This
enhanced kinetic behaviour can be understood as originating from a
downward shift in the $E_F$ relative to the charge neutrality point (CNP),

driven by the work function difference between RuCl$_3$ and MLG. To a
first approximation, the associated increased DOS (hole doping) at $E_F$
would greatly expand the availability of states to mediate interfacial ET.
In the Gerisher−Marcus framework, this increases the extent of energy
overlap between states in Ru(NH$_3$)$_6^{3+/(2+)}$ and MLG.

Finite-element simulations were performed using COMSOL Mul-
tiphysics (v.5.4) to simulate voltammetric responses with the Butler−
Volmer model and estimate standard electrochemical rate constants
($k^0$), as detailed in the Methods. Figure 1d illustrates the correlation
between $k^0$ and hBN thickness for RuCl$_3$ (solid-state acceptor) and
WSe$_2$ (solid-state donor), revealing a strong modulation in ET kinet-
ics with varying hBN thickness. Even when a 50-nm-thick hBN spacer
layer is used, RuCl$_3$ leads to a measurable increase in $k^0$ relative to the
electrochemical response measured in the absence of RuCl$_3$. In the
case of WSe$_2$, the effect diminishes beyond an hBN spacer thickness of
20 nm. In the absence of an hBN spacer, MLG−RuCl$_3$ achieves ET rates
comparable with those of pristine bulk graphite.

## Carrier density and fluorescence measurements

Raman spectroscopy and Hall measurements were used to quantify
doping as a function of hBN thickness[30] (Methods and Extended Data
Figs. 6 and 7). Figure 2a (top) compares the Raman G-peak positions of

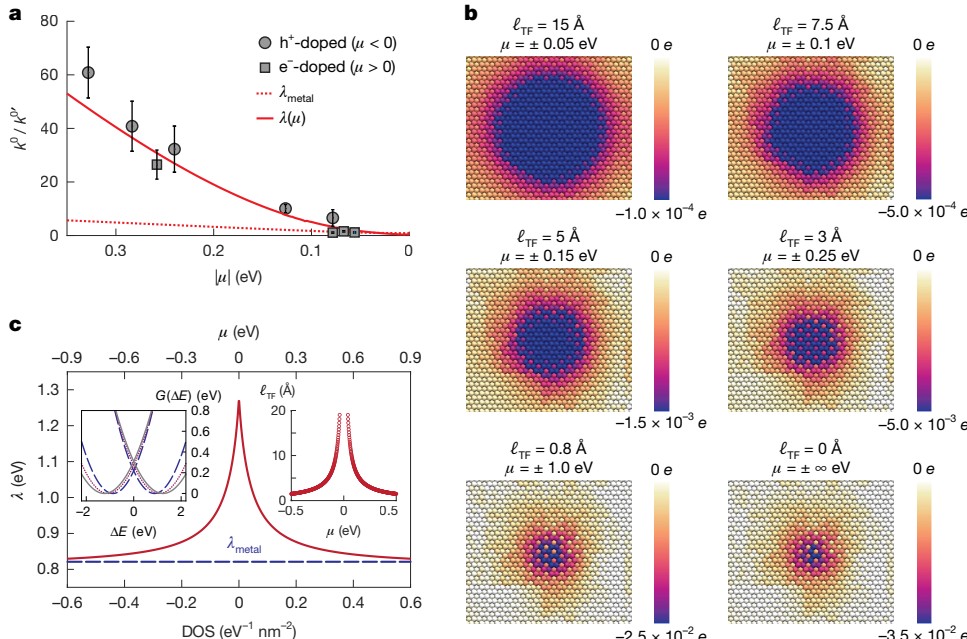

**Fig. 3 | DOS-dependent electrode polarization and charge-transfer kinetics.** **a**, Standard charge-transfer rate constants ($k^0$) as a function of $\mu$, normalized to the standard rate constant at undoped graphene ($k^{0'}$). Experimental data (symbols) are compared with the model using fixed $\lambda_{metal}$ (red dotted line) and $\lambda(\mu)$ derived from $\ell_{TF}$ (red solid line). Error bars indicate the s.d. for each $k^0$, where $n$ varies from 3 to 6. **b**, Simulated polarization response of the electrode upon switching the charge state of a redox ion. The redox ion is positioned at a fixed distance of 5 Å above the electrode surface in the $z$-direction. Polarization magnitude is visualized using an exponential colour scale. In the colour scale bar labels, $e$ is the elementary charge. **c**, Reorganization energy ($\lambda$) as a function of DOS and chemical potential ($\mu$). Left inset: free energy surfaces $G(\Delta E)$ of ET for $[Ru(NH_3)_6]^{3+/2+}$ redox couple for $\mu = 0$ eV (solid grey), 0.05 eV (dotted magenta) and 0.5 eV (dashed blue). Right inset: $\ell_{TF}$ versus $\mu$, calculated from DOS.

pristine MLG with those of MLG–RuCl$_3$, MLG–WSe$_2$ and MLG–hBN–RuCl$_3$ heterostructures across varying hBN thicknesses (Fig. 2a, bottom). The doping-induced shifts in the graphene G-peak position are stronger for RuCl$_3$ than WSe$_2$ and are seen to attenuate with increasing hBN thickness. Past work has shown a linear relationship between G peak position and charge carrier density[30], establishing a doping change of about $9 \times 10^{11}$ cm$^{-2}$ carriers per wavenumber shift. This correlation was used to deduce the graphene charge carrier density induced by RuCl$_3$ or WSe$_2$ heterolayers as a function of hBN spacer thickness.

When MLG and RuCl$_3$ are separated by hBN crystals thicker than 20 nm, we are unable to resolve further shifts in the G-peak. Given that we observed modulations in ET kinetics beyond 20 nm (Fig. 1d), we fabricated mesoscopic Hall bar devices and performed electronic transport measurements to measure these low carrier densities (Methods). Figure 2b presents the Hall resistance ($R_{xy}$) at 1.8 K for MLG–hBN–RuCl$_3$ devices with hBN thicknesses of 22, 43 and 79 nm. The slope of $R_{xy}$ as a function of magnetic field at low fields is inversely proportional to the carrier concentration, $n$, and indicates carrier type (positive, holes; negative, electrons), allowing for direct assessment of doping levels. Up to an hBN spacer thickness of 43 nm, RuCl$_3$ generates hole doping in MLG, whereas a negative slope is measured in the pristine MLG (no RuCl$_3$) region of the same device as well as in the MLG–hBN–RuCl$_3$ device comprising 79-nm-thick hBN, indicating electron doping.

Figure 2c plots $|n|$ in MLG as a function of hBN thickness, derived from Raman G-peak shifts and Hall measurements. These experimental values are compared with the doping expected from first-principles calculations[32] (black dashed line), which account for both the work function difference between MLG and RuCl$_3$, and the dielectric constants of the spacers. Although the model aligns very well with experimental data for hBN thicknesses of over 20 nm, doping levels exceed those predicted by theory for thinner hBN spacer layers. The enhanced doping observed in ultrathin hBN may be a result of valence band alignment

between hBN and RuCl$_3$, which activates a theoretically proposed defect-mediated charge transfer process[33]. To investigate this possibility of hBN-defect-mediated doping, we conducted fluorescence measurements on hBN–RuCl$_3$ heterostructures (Fig. 2e). Past work has shown that organic solvents enhance hBN defect fluorescence via solvent-defect charge transfer[34]. Using wide-field imaging (Methods), we observed that RuCl$_3$ strongly quenches the defect-based fluorescence in hBN (Fig. 2f). This result suggests that defect-mediated interactions may indeed contribute to the increased extent of MLG doping by RuCl$_3$, which is observed when hBN spacers less than 20-nm thick are used in MLG–hBN–RuCl$_3$ heterostructures.

## Effect of metallicity on the reorganization energy

By relating the hBN-modulated carrier densities (Fig. 2c) to shifts in the chemical potential $\mu$ (that is, $E_F$ at equilibrium) of MLG relative to its band structure, Fig. 3a compares experimentally measured outer-sphere standard ET rate constants (symbols) as a function of $\mu$ to theoretical models (lines). In our modelling (Methods and Extended Data Figs. 8 and 9), rate constants are normalized to those at the undoped graphene ($k^{0'}$), and are calculated at zero overpotential. Under these conditions, quantum capacitance effects on $\mu$ are negligible compared with the effects of solid-state doping.

First we consider an MHC model (red dotted line in Fig. 3a) in which the reorganization energy ($\lambda_{metal}$) is assumed to be a constant $\lambda = 0.82$ eV, as measured for this redox pair on metal (gold modified by self-assembled monolayer thiols) electrodes[35]. This MHC model accounts for the doping-dependent DOS but captures only the effect on interfacial ET of increasing the number of thermally accessible states in the electrode. The result is a prediction of very modest rate enhancement with DOS relative to that at the undoped graphene (about five to tenfold at extreme doping) that completely fails to replicate the experimental scaling.

Instead we find that the strongest contribution to the ET rate enhancement with increasing DOS is derived from an attenuation in the reorganization energy. This DOS dependence of $\lambda$ arises in response to changes in the dielectric screening within the electrode as a function of doping, and can be understood in terms of image charge interactions at the electrochemical interface. To explain this effect, Fig. 3b presents a series of simulation snapshots of the instantaneous discharging of an empty capacitor, displaying logarithmic maps of the induced charge distributions in response to a monovalent ion placed 5 Å away from the surface. The simulations were performed subject to a constant potential constraint, where electron–electron screening was modelled within TF theory[27,36,37]. The screening length, $\ell_{TF}$, is inversely proportional to the Fermi level, $\mu$, reflecting enhanced metallicity as the charge carrier density increases. As $|\mu|$ increases (corresponding to an increased metallicity and decrease in $\ell_{TF}$), the induced charge density becomes sharply localized, whereas a decrease in doping, or equivalently a decrease in metallicity, leads to a more diffuse charge distribution.

Next, in Fig. 3c we account for this $\ell_{TF}$ to model a reorganization energy, $\lambda(\mu)$, which is a function of DOS and $\mu$ using non-local dielectric continuum theory. Our model predicts a strong attenuation of $\lambda(\mu)$ with increasing metallicity (increasing DOS), converging to a value of 0.82 eV, consistent with $\lambda_{metal}$ (ref. 35). This trend arises from enhanced stabilization of the charge transfer transition state due to sharply localized polarization within the electrode. This manifests in the calculated free energy surfaces for the $Ru^{3+}$–$Ru^{2+}$ redox system at different values of $\mu$ (left inset of Fig. 3c). These calculations reveal a pronounced influence of electrode screening on ET kinetics: reduced $\mu$ monotonically increases the activation free energy ($\Delta G^{\ddagger}$), corresponding to the intersection of the reactant and product free energy surfaces. The right inset of Fig. 3c plots $\ell_{TF}$ versus $\mu$. The plot reveals a rapid decline in $\ell_{TF}$ as the system is doped away from the CNP ($\mu = 0$), directly linking doping-induced Fermi level shifts to screening efficiency. By explicitly including $\lambda(\mu)$—which decreases with rising DOS—our model achieves quantitative agreement with experimental rates (red line in Fig. 3a), establishing that electrode electronic structure governs $\lambda$ and by extension the activation free energy landscape. Thus, the predominant contributor to increasing ET rates with increasing DOS is the decrease in $\lambda$, and not the change in number of thermally accessible electron donor–acceptor states.

## Discussion and conclusions

Our experimental and theoretical results demonstrate the pivotal influence of the electrode electronic structure on interfacial ET kinetics, not only through the expected increase in the pre-exponential factor as DOS increases, but also through a significant impact on the reorganization energy itself. This paradigm shift challenges the traditional electrolyte-centric perspective of reorganization dynamics, compelling a revised activation free energy framework that explicitly integrates the influence of electrode electronic properties on $\lambda$. Our findings provide a more comprehensive mechanistic understanding of heterogeneous ET and establish foundational principles to guide the design and optimization of interfacial charge transfer in next-generation devices. We anticipate that this concept will be particularly relevant to photo-induced ET processes, which inherently involve semiconducting (thus low DOS) materials[38–40]. Future theoretical and experimental efforts should further unravel the interplay between electrode DOS, defect engineering and $\lambda$ to fully harness the potential of low-DOS materials in quantum technologies and energy applications.

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

## Methods

### Chemicals and materials

Natural Kish graphite crystals (Grade 300, 99.99% purity) were procured from Graphene Supermarket. Hexagonal boron nitride crystals were provided by T. Taniguchi and K. Watanabe, and were used as received. Large, flat crystals of $RuCl_3$ were grown by chemical vapour transport following the procedure detailed in a previous study[41]. Briefly, commercial $RuCl_3$ powder (Alfa Aesar, anhydrous, $Ru \geq 47.7\%$) was loaded into a quartz ampoule in an argon glovebox, sealed under dynamic vacuum, and heated in a two-zone furnace with a temperature gradient and ramp rates of 1 K per min. The resulting crystals were collected from the cold end and stored in an argon-filled glovebox.

Si/SiO$_2$ wafers (0.5-mm-thick, 285 nm SiO$_2$) and polydimethylsiloxane stamps (PDMS) were obtained from NOVA Electronic Materials and MTI Corporation, respectively. Sn/In alloy (Custom Thermoelectric), poly(bisphenol-A carbonate), hexaammineruthenium(III) chloride (98%) and potassium chloride (>99%) were purchased from Sigma-Aldrich. Sulfuric acid (ACS grade, >95–98%, Thermo Fisher Scientific) was used as received. All aqueous electrolyte solutions were prepared using type I water (EMD Millipore, 18.2 MΩ cm). Solid KCl was added as a supporting electrolyte in $Ru(NH_3)_6^{3+}$ solution to a final concentration of 100 mM.

### Sample fabrication

Graphene and hBN flakes were mechanically exfoliated onto SiO$_2$ (285 nm)/Si substrates from bulk crystals using Scotch tape[42]. Exfoliated flakes on SiO$_2$/Si chips were identified by optical microscopy (Laxco LMC-5000). MLG flakes were distinguished by their approximately 7% optical contrast in the green channel[14,43] and further verified by Raman spectroscopy (HORIBA LabRAM Evo). Extended Data Fig. 1 shows a representative optical contrast of around 7% in the green channel for MLG and about 14% for bilayer graphene. The thickness of hBN flakes was determined by atomic force microscopy (Park Systems NX10) (Extended Data Fig. 1c,d).

α-RuCl$_3$ crystals were exfoliated in an argon-filled glovebox onto SiO$_2$ (90 nm)/Si substrates to prevent degradation. Precise thickness control was not required, as even a single monolayer of α-RuCl$_3$ is sufficient to induce substantial hole doping in graphene[30,41]. Instead, emphasis was placed on selecting flakes smaller than the hBN to ensure complete encapsulation, and flatness was prioritized to minimize strain during stacking. Suitable flakes were identified with an optical microscope (Nikon) within the glovebox.

We selected the multilayer system comprising graphene, hBN, RuCl$_3$ and WSe$_2$ due to their complementary characteristics. Graphene offers a tunable and well-defined electronic platform, whereas hBN serves as an inert spacer that allows precise control of doping. The RuCl$_3$ and WSe$_2$ layers function as stable charge-transfer dopants, modulating graphene's electronic properties without affecting its structural integrity. Together, these materials enable systematic tuning of interfacial doping while preserving the overall structural quality of the heterostructure. MLG–hBN–RuCl$_3$ heterostructures were assembled by a dry-transfer technique on a temperature controlled stage (Instec), equipped with an optical microscope (Mitutoyo FS70) and micromanipulator (MP-285, Sutter Instrument) in an argon glovebox. A poly(bisphenol-A carbonate) film on a PDMS stamp was used to pick up a RuCl$_3$ flake within 30 min of exfoliation to minimize moisture exposure, which could compromise its doping efficacy[41]. The picked RuCl$_3$ flake was then capped with an hBN flake (3–180 nm thick), followed by MLG, partially overlapping the RuCl$_3$ to leave a segment of graphene without RuCl$_3$. A thick graphite flake (10–100 nm) was finally transferred to partially overlap the graphene, providing electrical contact with the heterostructure. The poly(bisphenol-A carbonate) film was delaminated from the PDMS stamp and placed onto a clean SiO$_2$/Si chip. Electrical contacts with graphite were subsequently established using Sn/In microsoldering[14].

### SECCM measurements

Single-channel SECCM nanopipettes were fabricated from quartz capillaries (0.7 mm inner diameter, 1 mm outer diameter; Sutter Instrument) using a laser puller (P-2000, Sutter Instrument) with the following parameters: heat = 700, filament = 4, velocity = 20, delay = 127, and pull = 140. This procedure yielded pipettes with orifice diameters of 600–800 nm, as confirmed by bright-field transmission electron microscopy (TEM; Extended Data Fig. 3). Each nanopipette was filled with an electrolyte solution containing the redox species of interest and equipped with a silver wire coated with AgCl, serving as a quasi-reference or counter electrode.

Scanning electrochemical cell microscopy experiments were performed using a Park NX10 SICM module. The nanopipette was positioned above the sample using an optical microscope and approached the surface at 100 nm s$^{-1}$ until meniscus contact was detected by a current increase above 3 pA. During approach, a −0.5 V bias was applied to facilitate diffusion-limited reactions. Cyclic voltammograms were recorded at multiple locations by sweeping the potential at 100 mV s$^{-1}$ between −0.6 V and 0 V, with the half-wave potential, $E_{1/2}$, defined as the potential at which $i = i_\infty/2$, where $i_\infty$ represents the diffusion-limited current plateau. [$Ru(NH_3)_6^{3+/2+}$] was chosen as the redox couple because it has well-characterized, reversible, outer-sphere ET with no detectable adsorption on graphite electrodes, as confirmed by in situ Raman spectroscopy[14,21]. This ensures that the measured kinetics are sensitive to the electronic properties of the electrode while avoiding complications from surface-specific reactions.

Measurements were conducted on multiple independently fabricated devices, each featuring a distinct hBN thickness and comprising regions of evaporated gold as well as MLG with and without RuCl$_3$. Notably, the thickness data for 0- and 3-nm-thick hBN were measured on the same device, as were the data for 77- and 93-nm-thick hBN. For devices without hBN, RuCl$_3$ and WSe$_2$ are sensitive to air exposure, so the entire MLG was used to encapsulate them and, consequently, no isolated MLG regions were available.

For each device, we recorded 1–2 voltammetric cycles at multiple spatially separated positions to ensure reproducibility and capture local variability. Voltammetric data from each MLG position, including multiple cycles, were binned and individually fitted to COMSOL simulations to extract $k^0$ values. The gold regions served as an internal reference, with their data fitted using a reversible rate constant of 0.5 cm s$^{-1}$ to account for any variations $E^0$. This yielded multiple $k^0$ values per device. The values plotted in Fig. 1d represent averages across these measurements, with error bars indicating the standard deviation. All extracted $k^0$ values are provided in Extended Data Table 1. We find that the enhancements in $k^0$ observed here far exceed those predicted by MHC theory, consistent with other studies that have reported similar limitations of the MHC framework[18–20,44–46].

Although our devices were measured on the same day as fabrication, a 4–5 h interval was required for device assembly—including stacking, making electrical contacts, and transfer to the SECCM measurement substrate—which may have contributed to the slower observed rates than reported in literature. In this context, having MLG regions without RuCl$_3$–WSe$_2$ on the same device provides a robust baseline to reliably study the relative enhancement in ET kinetics induced by these dopants.

Past contact angle studies on graphene report modest changes (from 105° to 90°) over several minutes[47], which is significantly longer than our measurement timescale (<10 s). Molecular dynamics simulations show that increasing surface charge reduces wettability[48], suggesting that electrowetting effects should be even weaker in doped graphene. Electrowetting experiments on highly oriented pyrolytic graphite in 0.1 M KF over a potential window of 0 to −2.0 V versus Ag/AgCl revealed negligible effects[49], consistent with our experimental conditions (0.1 M KCl, 0 to −0.7 V versus Ag/AgCl). Our cyclic voltammetry signals remained stable, and microscopy before and after testing confirmed

no detectable morphological changes. These observations indicate that electrowetting does not significantly affect our measurements.

## Finite-element simulations

Finite-element simulations of steady-state cyclic voltammograms were performed using COMSOL Multiphysics (v.5.6)[50], following a similar approach outlined in previous works[14,21]. The nanopipette geometry was modelled in a 2D axisymmetric configuration (Extended Data Fig. 2), with droplet radii assumed equal to the pipette aperture, consistent with past studies[14,21,51]. The pipette radius, $a_s$, and taper angle, $\theta_s$, were determined from TEM images (Extended Data Fig. 3). A survey of multiple nanopipettes prepared under identical conditions revealed that the taper angles are highly consistent ($14.1 \pm 0.3°$), whereas the aperture sizes have a modest distribution ranging from 600 to 800 nm.

Mass transport of redox species was modelled using the 'Transport of diluted species' and 'Electrostatics' modules, solving the steady-state Nernst–Planck equation:

$$D_i\left(\frac{\partial^2 c_i}{\partial r^2} + \frac{1}{r}\frac{\partial c_i}{\partial r} + \frac{\partial^2 c_i}{\partial z^2}\right)$$
$$= -\frac{z_i F c_i D_i}{RT}\left(\frac{\partial^2 \phi}{\partial r^2} + \frac{1}{r}\frac{\partial \phi}{\partial r} + \frac{\partial^2 \phi}{\partial z^2}\right); \quad 0 < r < r_s, \ 0 < z < l \tag{1}$$

where $r$ and $z$ represent the coordinates parallel and normal to the sample surface, respectively; $F$ is the Faraday constant; and $r_s$ and $l$ denote the width and height of the simulation space, respectively. The height $l = 30$ m was set to exceed the nanopipette aperture, ensuring boundary effects were negligible. The meniscus was modelled as a cylindrical droplet (height, $h$), consistent with the hydrophobic interaction of water on graphite (contact angle 90°)[14,21]. The electroactive radius, $a$, is set equal to the nanopipette radius $a_s$, in agreement with previous studies[14,21]. The variables $c_i$, $z_i$ and $D_i$ represent the concentration, charge number and diffusion coefficient, respectively, of either the oxidized ($c_O$) or the reduced ($c_R$) form. The electric potential $\phi$ in solution is determined by solving the Poisson equation:

$$\frac{\partial^2 \phi}{\partial r^2} + \frac{1}{r}\frac{\partial \phi}{\partial r} + \frac{\partial^2 \phi}{\partial z^2} = -\frac{\sum_i z_i F c_i}{\varepsilon \varepsilon_0}; \quad 0 < r < r_s, \ 0 < z < l \tag{2}$$

where $\varepsilon = 80$ is the dielectric constant of the solvent (water), and $\varepsilon_0$ is the vacuum permittivity. The terms $c_i$ and $z_i$ in equation (2) include the ions of the supporting electrolyte (0.1 M KCl) in addition to the redox-active species $c_O$ and $c_R$. The rate of the heterogeneous electron-transfer reaction is governed by the Butler–Volmer equations:

$$k_{red} = k^0 e^{-\alpha\frac{F}{RT}(V_{app}-E_0)} \tag{3}$$

$$k_{ox} = k^0 e^{(1-\alpha)\frac{F}{RT}(V_{app}-E_0)} \tag{4}$$

where $k^0$ is the standard rate constant, $\alpha$ is the transfer coefficient, $E_0$ is the standard potential and $V_{app}$ is the applied electrochemical potential. For the simulation of $Ru(NH_3)_6^{3+/2+}$, only the oxidized form ($c_O$) is initially present in the solution. The flux is considered zero except at the contact surface. The general boundary conditions are given as follows:

$$c_O = c_O^*, \ c_R = c_R^* = 0; \quad 0 < r \le r_s, \ z = l \ \text{(bulk)} \tag{5}$$

$$\frac{\partial c_i}{\partial n} = 0; \quad 0 < z \le h, \ r = a_s;$$
$$h < z < l, \ r = a + (z-h)\tan(\theta_p) \ \text{(no flux)} \tag{6}$$

$$J_O = -J_R = k_{red}c_O - k_{ox}c_R; \quad 0 < r \le a_s, \ z = 0 \ \text{(sample surface)} \tag{7}$$

where $J_O$ and $J_R$ represent the inward flux of the oxidized and reduced forms, respectively, and $c_O^*$ and $c_R^*$ are the bulk concentrations. The $\frac{\partial c_i}{\partial n}$ term is the normal derivative of the concentration. The potential drop across the Helmholtz layer is implemented by defining the surface charge density, $\sigma$, at the sample surface:

$$\sigma = \frac{(V_{dl}-\phi)\varepsilon_H\varepsilon_0}{d_H}; \quad 0 < r \le a_s, \ z = 0 \tag{8}$$

where $\varepsilon_H = 6$ and $d_H = 0.5$ nm are the dielectric constant and thickness of the Helmholtz layer, respectively, yielding the double-layer capacitance $C_{dl} = 10$ μF cm$^{-2}$; $V_{dl}$ is the corresponding double-layer potential relative to the charge neutrality point. The steady-state current was calculated by integrating the total flux of the reactants ($J_O$) normal to the sample surface:

$$i = 2\pi F \int_0^{a_s} J_O r \, dr \tag{9}$$

The diffusion coefficients $D_O$ and $D_R$ for the $Ru(NH_3)_6^{3+/2+}$ couple were set to $8.43 \times 10^{-6}$ cm$^2$ s$^{-1}$ and $1.19 \times 10^{-5}$ cm$^2$ s$^{-1}$, respectively; $\alpha = 0.5$ was used for all simulations, consistent with previous studies on graphene thin films[14,21]. Our observed rates for doped MLG are ≤0.02 cm s$^{-1}$, and for graphite approximately 0.03 cm s$^{-1}$, indicating that ET remains primarily kinetically controlled within the experimental window. $E_0$ was determined from electrochemically reversible voltammograms obtained on gold electrodes immediately before the experiments on graphene.

To extract the standard rate constant $k^0$ from experimental voltammograms, we performed finite-element simulations across a range of $k^0$ values and computed residuals between simulated and experimental data via sigmoidal fitting. For each simulation, the coefficient of determination ($R^2$) was calculated using least-squares minimization, with the optimal $k^0$ corresponding to the maximum $R^2$ (minimal residuals). This protocol is illustrated in Extended Data Fig. 4, where $R^2$ values for simulated rates are plotted alongside representative voltammograms.

## Quantum capacitance

Quantum capacitance ($C_q$) is a material-specific capacitance that arises from the DOS at the $E_F$ in low-dimensional materials such as graphene[14,52]. When an electric potential ($V_{app}$) is applied across a solid–solution interface, an electric double layer (EDL) forms at the surface to screen the excess charge[5,53]. In low-dimensional systems such as MLG, this EDL functions not only as a charge screening layer but also as an electrostatic 'gate', shifting the Fermi level and dynamically altering the material's carrier concentration through electron or hole doping. In the case of MLG, applying $V_{app}$ results in two potential contributions: $V_q$, which is the potential change due to $C_q$, and represents the shift in the chemical potential; and $V_{dl}$, the potential drop across the double layer itself. These two components are related by:

$$V_{app} = V_q + V_{dl} \tag{10}$$

The EDL capacitance, $C_{dl}$, in an aqueous solution, is estimated around 10 μF cm$^{-2}$, assuming a compact layer capacitance with little dependence on ionic strength[54]. The diffuse-layer capacitance is >100 μF cm$^{-2}$ in 0.1 M KCl solution[53] and can be neglected. The total capacitance $C_{total}$ combines $C_q$ and $C_{dl}$ in series:

$$\frac{1}{C_{total}} = \frac{1}{C_q} + \frac{1}{C_{dl}} \tag{11}$$

**Calculating quantum capacitance for MLG.** Quantum capacitance is fundamentally connected to the DOS at the Fermi level, which depends on the material's band structure. For MLG, the quantum capacitance $C_q$ can be expressed as:

$$C_q = e^2 \frac{dn}{dV_q} \tag{12}$$

where $e$ is the elementary charge, and $\frac{dn}{dV_q}$ represents the rate of change in carrier concentration $n$ with respect to $V_q$. Under the two-dimensional free-electron gas model, considering graphene's linear DOS near the Dirac point[55], this relation simplifies to

$$C_q = \frac{2e^2 k_B T}{\pi \hbar^2 \upsilon_F^2} \ln\left[ 2\left( 1 + \cosh\left( \frac{eV_{ch}}{k_B T}\right)\right)\right], \tag{13}$$

where $\hbar$ is the reduced Planck constant, $k_B$ is the Boltzmann constant, $\upsilon_F \approx c/300$ is the Fermi velocity of Dirac electrons and $V_{ch} = E_F/e$ is the graphene potential. At the Dirac point, where carrier concentration $n$ is minimal, $C_q$ approaches zero. At $T = 300$ K, the channel potential can be written as:

$$eV_{ch} = \mu + eV_{app}. \tag{14}$$

Assuming constant charge, the relationship between $C_q$ and $C_{dl}$ is

$$\frac{C_q}{C_{dl}} = \frac{V_{dl}}{V_q}. \tag{15}$$

Substituting $C_{dl} = 0.1$ F m$^{-2}$ gives

$$V_{dl} = 10\, C_q V_q. \tag{16}$$

This leads to the relation between $V_q$ and the applied potential $V_{app}$:

$$V_{app} = (1 + 10\, C_q) V_q. \tag{17}$$

Finally, substituting equation (13) into equation (17) yields

$$\frac{V_q}{V_{app}} = 1 + 10 \times \frac{2e^2 k_B T}{\pi \hbar^2 \upsilon_F^2} \ln\left[ 2\left( 1 + \cosh\left( \frac{eV_{ch}}{k_B T}\right)\right)\right]. \tag{18}$$

This expression provides $V_q/V_{app}$ as a function of $V_{app}$, from which $V_{dl}(V_{app})$ is extracted for different values of $\mu$ and incorporated into our COMSOL simulations to systematically account for quantum capacitance effects. Extended Data Fig. 5 shows the ratio $V_q/V_{app}$ as a function of applied potential at 300 K. The inset presents the corresponding $C_q$ as a function of $V_{app}$.

## Raman spectroscopy measurements

**Sample preparation.** The heterostructures used in this study were prepared using a dry transfer method in an argon-filled glovebox. A polymeric stamp consisting of poly(bisphenol-A carbonate) on PDMS was used to pick up a thin layer of hBN, thickness < 5 nm, which was then used to pick MLG and another hBN flake comprising steps of multiple thickness, ensuring that the multilayer hBN fully covered the MLG. The entire stamp was then placed onto freshly exfoliated RuCl₃ on a Si/SiO₂ substrate (90-nm-thick SiO₂), and the PDMS was gently lifted at 160 °C, leaving behind the poly(bisphenol-A carbonate). The resulting structure consisted of Si/SiO₂-RuCl₃-multilayer hBN–MLG-thin hBN–PC. The thin hBN layer—large enough to cover the entire heterostructure—served as a capping layer to protect the stack from solvents. The poly(bisphenol-A carbonate) was then dissolved in chloroform for 15 min, leaving the heterostructure device ready for Raman measurements. Notably, doping is localized to the graphene in contact with α-RuCl₃, forming an atomically sharp lateral junction[30,41]. During fabrication, defects such as gas bubbles trapped between layers or non-uniform strain distributions may modulate the coupling between the layers locally. This coupling controls charge transfer between graphene and α-RuCl₃, as seen in the distance dependence introduced by hBN spacers. It is therefore crucial to freshly exfoliate RuCl₃ just before stacking to avoid contamination, which could decouple the layers.

**Spectra acquisition and analysis.** Confocal Raman spectra were collected using a Horiba Multiline LabRam Evolution system with a 532 nm laser and 0.4–3 mW power, using either a 600- or 1,800-grooves-per-millimitre grating. Spectra were typically recorded with acquisition times of 5–10 s and 3–5 accumulations. The G peak position in the Raman spectra can be linearly correlated with the doping levels in graphene, particularly when modulated by the electric field effect[30,41]. In the MLG–RuCl₃ device, the G peak is blue-shifted by more than 25 cm$^{-1}$ relative to the region without RuCl₃, indicating doping of approximately $2.5 \times 10^{13}$ holes per cm², consistent with previous studies[30,41]. Such studies have shown that the G and 2D peak shifts in graphene vary with doping induced by the electric field effect[30,41], and the G peak shift has a quasi-linear dependence on doping. Averaging the slopes of the G peak shift versus carrier density across these studies yields a value of approximately $9 \times 10^{11}$ cm$^{-2}$ carriers per wavenumber shift in the G peak position. Voigt profiles are used to fit the peaks, accounting for the Lorentzian nature of the phonons and the Gaussian instrumental resolution[30,41]. A constant background is subtracted from each spectrum before fitting the peaks. The 2D peak position provides a sensitive probe of strain and morphological changes. Extended Data Fig. 6 shows negligible shifts in the 2D peak for MLG with and without hBN/RuCl₃, across varying hBN thicknesses, thereby confirming the absence of significant geometric alteration.

**First-principles modelling of doping in MLG/hBN/RuCl₃ heterostructures.** The MLG/hBN/RuCl₃ heterostructure was modelled as a parallel-plate capacitor following Bokdam et al.[32], with the Fermi level shift given by:

$$\Delta E_F(E_{ext}) = \pm \frac{\sqrt{1 + 2D_0\alpha\left(\frac{d}{\kappa}\right)^2 |e(E_{ext} + E_0)|} - 1}{D_0\alpha d/\kappa} \tag{19}$$

Here, $\alpha = \frac{e^2}{\varepsilon_0 A} = 34.93$ eV Å$^{-1}$ (where $A = 5.18$ Å² is the area of the graphene unit cell, and $\varepsilon_0$ is vacuum permittivity), $D_0 = 0.102$ eV$^{-2}$ per unit cell (slope of MLG DOS), $d$ is the dielectric spacer thickness, $\kappa$ is relative permittivity and $E_{ext}$ is the external electric field. $E_0$ accounts for any built-in electric field or doping potential.

**Experimental validation of defect-mediated doping.** To resolve the anomalous doping in thin hBN heterostructures (<20 nm), we fabricated devices with alternating hBN-supported (MLG/hBN/RuCl₃) and suspended MLG regions using approximately 4-nm-thick hBN, as shown in Extended Data Fig. 7 (MLG/air/RuCl₃). Doping levels were calculated using equation (19) and measured via Raman G-peak shifts (Extended Data Table 2).

The suspended region shows good agreement between theory and experiment. In contrast, the hBN-supported region has 21% higher doping than predicted. This discrepancy, along with the thickness-dependent deviations shown in Fig. 2e, suggests that defect states in hBN do contribute further charge transfer beyond classical capacitive coupling.

## Liquid-activated fluorescence measurements

The sample was mounted on a Nikon Ti-E inverted fluorescence microscope equipped with a 100× oil-immersion objective lens (CFI Plan Apochromat $\lambda$ 100×, NA = 1.45). Intensities in Fig. 3c were captured under 561 nm laser excitation (OBIS 561LS, Coherent, 165 mW) with an exposure time of 6 ms. Emission was collected after a band-pass filter (ET605/70m, Chroma).

## Nanofabrication of Hall measurement devices

All device fabrication was performed in the Marvell Nanofabrication Laboratory. Electron beam lithography (CRESTEC CABL-UH system), with an A6 950 poly(methyl methacrylate) resist, was used to define the

electrode and contact regions. Reactive ion etching (SEMI RIE system) exposed the graphene edges in the hBN–graphene–RuCl$_3$ heterostructure through a sequence of plasma treatments: 15 s of O$_2$ plasma to remove surface residues; 40 s of SF$_6$/O$_2$ plasma to etch through hBN and reveal the graphene; and a final 15 s of O$_2$ plasma to eliminate etching by-products. Immediately following etching, Cr/Au (20/120 nm) was deposited by thermal evaporation (NRC Evaporator) at rates of 0.5 Å s$^{-1}$ for Cr and 2–4 Å s$^{-1}$ for Au to form electrical contacts and bonding pads. After lift-off, a second electron beam lithography step defined an etch mask for shaping the heterostructure into a Hall bar geometry, minimizing longitudinal and transverse resistance mixing. The poly(methyl methacrylate) mask was retained after fabrication to protect RuCl$_3$ from environmental degradation. Device integrity was verified by measuring resistance between electrical contacts using a lock-in amplifier (Stanford Research SR830) on a probe station. Functional devices were wire bonded (TPT HB05) to 16-pin ceramic dual inline packages for subsequent measurements in a Quantum Design PPMS.

**Theoretical preliminaries**

The [Ru(NH$_3$)$_6$]$^{2+/3+}$ redox couple is known to undergo ET through an outer-sphere mechanism, where the thermal fluctuations of long-ranged electrostatic interactions with the environment play a central role in mediating the dynamics[56]. In the weak-coupling (non-adiabatic) regime, the rate of interfacial reduction is well-described by the golden-rule expression[6,57],

$$k_{red}(E_F) = \frac{2\pi}{\hbar}|V|^2 \int_{-\infty}^{\infty} D(E) f_{E_F}(E) \langle \delta(\Delta E - E) \rangle_{ox} dE \quad (20)$$

$$= \frac{2\pi}{\hbar}|V|^2 \int_{-\infty}^{\infty} D(E) f_{E_F}(E) p_{E_F}^{(ox)}(E) dE \quad (21)$$

Where $V$ is the electronic coupling between the two redox states (assumed to be small); $D(E)$ is the electrode's DOS; $f_{E_F}(E)$ is the Fermi–Dirac distribution, centred at the electrode's Fermi level, $E_F$;

$$f_{E_F}(E) = \frac{1}{1 + e^{(E-E_F)/k_B T}} \quad (22)$$

and $p_{E_F}^{(ox)}(\Delta E)$ is the equilibrium probability distribution of the vertical energy gap between the two charge transfer states, evaluated in the oxidized state. Under an assumption of linear dielectric response, the rate is appropriately computed with Marcus theory, where the energy gap obeys Gaussian statistics and the free energy surfaces of ET are parabolic[6,7],

$$-\ln p_{E_F}^{(ox)}(E) = \frac{(E - E_F + \lambda)^2}{4\lambda k_B T} + \frac{1}{2}\ln[4\pi k_B T] \quad (23)$$

An assumption of chemical equilibrium is made, such that the intersection of the Marcus curves is aligned with the electrode's Fermi level[58]. The reorganization energy, $\lambda$—a critical determinant of the activation free energy—quantifies the reversible work required to deform the equilibrium solvation environment of a redox species into that of its counterpart without ET. Equivalently, $\lambda$ represents the energy dissipated during a vertical transition, reflecting the solvent and electrode polarization response to instantaneous charge redistribution.

**The effect of electrode metallicity on the reorganization energy.** In outer sphere redox reactions, the electrostatic potential at the interface is critical in determining the ET rate[59,60]. As the Fermi level is shifted, the change in DOS renormalizes the material's electrostatic interactions, reflecting variations in the material's dielectric response function due to an altered number of charge carriers that can respond to external fields. Insight into this effect can be gained through simple models of electronic screening such as TF theory[61,62] parametrized by a screening

length, $\ell_{TF}$, which sets the scale of exponential decay of electrostatic interactions in the material, and interpolates between a perfect metal ($\ell_{TF} = 0$) and an insulating material ($\ell_{TF} \to \infty$). The screening length is closely related to a material's low-energy electronic structure. In two dimensions, and for $k_B T \ll E_F$, which is typically the case for the energy scale of valence electrons, $\ell_{TF}$ is[63]

$$\ell_{TF} = \frac{\epsilon^{(el)}}{2\pi e^2 D(E_F)} \quad (24)$$

where $n$ is the charge density of the material. Marcus derived dielectric continuum estimates of the reorganization energy for charge transfer with a perfect conductor $\ell_{TF} = 0$ (ref. 7),

$$\lambda = -\frac{\delta q^2}{4z_0}\left(\frac{1}{\epsilon_\infty^{(sol)}} - \frac{1}{\epsilon^{(sol)}}\right) + \lambda_B, \quad (25)$$

where $\delta q$ is the amount of transferred charge, $\epsilon_\infty^{(sol)}$ and $\epsilon^{(sol)}$ are the electrolyte's optical and static dielectric constants respectively, $z_0$ is the separation from the electrochemical interface, and $\lambda_B$ is the bulk contribution to the reorganization energy, which dominates when far away from the interface. The $(1/\epsilon_\infty^{(sol)} - 1/\epsilon^{(sol)})$ term, known as the Pekar factor[64], is a measure of the free energy difference between a charge exclusively solvated by the medium's fast (electronic) degrees of freedom, and that of a charge fully stabilized by the polarization field's nuclear and electronic degrees of freedom.

The electrostatic potential at dielectric discontinuities can be obtained by solving Poisson's equation with appropriate boundary conditions at the interface. If the two media that make up the interface are complex materials with some degree of unbound charge, the dielectric constant is replaced by a dielectric response function $\epsilon(\mathbf{r})$, and the non-local Poisson equation reads,

$$\nabla \cdot \int d^3\mathbf{r}' \epsilon_\alpha(\mathbf{r} - \mathbf{r}')\nabla\phi(\mathbf{r}') = -\rho_\alpha(\mathbf{r}) \quad (26)$$

where $\alpha$ labels the medium, and $\rho_\alpha(\mathbf{r})$ is the charge density in medium $\alpha$. The boundary conditions to be satisfied are the continuity of the potential and the electric displacement field across the interface. For a point charge near a perfect conductor, the potential energy due to the conductor's polarization in response to the external field can be mapped to the interaction between the real charge and an opposite-sign 'image' charge placed symmetrically inside the metal. The electrode contribution to the reorganization energy in equation (25) directly corresponds to the electrostatic interaction of a point charge $\delta q$ with its image, weighted by the Pekar factor. It provides reasonable estimates of the reorganization energy at electrodes that approach the behaviour of an ideal conductor, but becomes inaccurate away from this limit. We have recently developed a formalism[25]—building on previous developments[65–67]—that describes how the solvent reorganization energy changes as a function of the electrode's metallicity in the context of TF theory. If we consider the electrostatic boundary-value problem of a charge $q$ embedded in a dielectric in contact with a material with finite TF screening, Poisson's equation can be solved making use of Fourier-Bessel transforms[65,66]. The end result is an expression that encodes the interaction of the point charge $q$ with the induced charge density in the electrode, as well as the self energy of the induced charge density. Although the Fourier–Bessel transform of the potential cannot be analytically inverted, an analogy can be made to the method of images to write the electrostatic potential energy of this system as the effective interaction of $q$ with a modified image charge at $-z_0$,

$$U(z_0, \ell_{TF}) = \frac{q^2 \xi_{\ell_{TF}}(z_0, \epsilon^{(sol)})}{4\epsilon^{(sol)} z_0} \quad (27)$$

where we have defined the image charge scaling function, $\xi_{\ell_{TF}}(z_0, \epsilon^{(sol)})$, which informs on the value (with respect to $q$) of this fictitious image charge as a function of screening length $\ell_{TF}$, and the dielectric constants of both media. It smoothly interpolates between the electrostatics at the boundary of an ideal conductor, and an insulator. We see that, at a fixed $z_0$, the TF screening length in the electrode takes the image charge from $-q$ to $\xi_\infty(z_0, \epsilon^{(sol)})q$. The screening-dependent image potential results in a modified reorganization energy,

$$\lambda(\ell_{TF}) = \frac{\delta q^2}{4z_0}\left(\frac{\xi_{\ell_{TF}}(z_0, \epsilon_\infty^{(sol)})}{\epsilon_\infty^{(sol)}} - \frac{\xi_{\ell_{TF}}(z_0, \epsilon^{(sol)})}{\epsilon^{(sol)}}\right) + \lambda_B \quad (28)$$

The term in parenthesis in equation (28) can be identified as a generalization of the Pekar factor, extended to describe the modulation of image interactions at the surface of a TF electrode.

**Image interactions in hole-doped MLG.** The low-energy band structure of graphene can be described analytically, obeying the well-known linear dispersion relation characteristic of massless Dirac fermions[68]. This in turn results in a linear electronic DOS,

$$D(E) = \frac{2}{\pi\hbar^2 v_F^2}|E| \quad (29)$$

where $v_F \approx 10^6$ ms$^{-1}$ is the Fermi velocity, and the electronic energy $E$ is measured from the Dirac point. Under a low-temperature approximation, graphene's charge density is[69],

$$\rho(E_F) = \text{sgn}(E_F)\frac{E_F^2}{\pi\hbar^2 v_F^2} \quad (30)$$

leading to an explicit relationship between $\ell_{TF}$ and the Fermi level[70],

$$\ell_{TF}(E_F) = \frac{\epsilon^{(el)}\hbar^2 v_F^2}{4e^2|E_F|}. \quad (31)$$

The separation of the redox ion from the electrode, $z_0$, must be chosen judiciously. Given the outer-sphere nature of the reaction, it must account for the structure of the interface, including an adlayer of water molecules on the surface of the electrode that are tightly bound and held together by a hydrogen-bonding network[71,72], as well as the inner coordination environment and the outer solvation shell of the redox species. On the other hand, as the diabatic coupling term typically decays exponentially with separation $V \approx V_0 e^{-z_0/z_{ref}}$, the rate will be dominated by the distance of closest approach to the electrode, so an estimated lower bound of the separation should always be chosen. A distance of $z_0 = 6$ Å was set on the basis of these considerations.

With an understanding of how the reorganization energy is modified in response to doping, the rate of electro-reduction may be estimated as:

$$k_{red}(E_F) = \frac{2|V|^2}{\hbar^3 v_F^2\sqrt{\pi^3\lambda(E_F)k_BT}}\int\frac{|E|e^{-\frac{(E-E_F-\lambda(E_F))^2}{4k_BT\lambda(E_F)}}}{1+e^{(E-E_F)/k_BT}}dE. \quad (32)$$

In keeping with the non-adiabatic limit that makes this treatment valid, we assume that the electronic coupling $|V|$ remains small regardless of the degree of doping. In fact, we take this factor to be roughly constant such that it approximately cancels when taking the ratio with respect to some reference, for instance the CNP, $k(E_F)/k_{CNP}$. This allows us to assess the behaviour of the rate as a function of doping without direct knowledge of $|V|$.

As noted earlier, $\lambda$ arises from solvation energy changes during instantaneous charge transfer between redox species and the electrode.

These changes are stabilized exclusively by fast solvent polarization modes, quantified by the optical dielectric constant $\epsilon_\infty^{(sol)}$. In water, the static dielectric constant vastly exceeds the optical dielectric constant, rendering the second term in equation (28) negligible compared with the first. Therefore, to a reasonable approximation, the behaviour of the reorganization energy will closely resemble the image potential of a charge interacting with the electrode only through the optical dielectric constant, close to 1 in water.

**Adiabatic versus non-adiabatic outer-sphere ET in $[Ru(NH_3)_6]^{3+/2+}$**
The question of whether outer-sphere ET in $[Ru(NH_3)_6]^{3+/2+}$ proceeds adiabatically or non-adiabatically remains an area of active debate in electrochemistry. We assume that interfacial ET for $[Ru(NH_3)_6]^{3+/2+}$ falls within the non-adiabatic regime, which contrasts previous work by Liu and co-workers[73], who proposed adiabatic ET for this redox couple at graphene electrodes. Several observations support our assumption.

First, we find that the rate is highly sensitive to electronic properties of the electrode, which is a distinctive characteristic of non-adiabatic ET. Furthermore, there is contrasting experimental evidence to that presented with regard to the adiabaticity of outer sphere ET in the $[Ru(NH_3)_6]^{3+/2+}$ redox couple at graphene electrodes[73]. The alluded work posits adiabatic ET on the basis of the assumption that increasing graphene layers equates to increasing tunnelling distance. However, this perspective may neglect the role of graphene's intrinsic electronic states, which actively participate in ET, making the simplified tunnelling argument inadequate, as we have illustrated in this work. Conversely, studies using hBN as a true inert spacer on graphite electrodes indeed demonstrate an exponential decrease in ET rates with increasing spacer thickness, consistent with non-adiabatic tunnelling processes[74,75]. The pronounced dependence of ET kinetics on the electrode's DOS observed here and in past studies further supports this interpretation[14,21]. Finally, ET for the $[Ru(NH_3)_6]^{3+/2+}$ system has been reported to occur near the outer Helmholtz plane, where the redox species is separated from the electrode surface by a structured solvent layer. Experimental evidence, including negative activation volumes, indicates ET through solvated species rather than direct electrode contact, implying weak electronic coupling[76]. Furthermore, the work of Nazmutdinov and colleagues supports our assumption[77]. They argue that 'for all amine complexes residing outside of the compact layer, ET proceeds in a diabatic limit, which originates mostly from a strong localization of the molecular acceptor orbitals on the central atoms'; this directly aligns with our picture of weak electronic coupling mediated through the solvent layer. Collectively, these observations align with our assumption of non-adiabatic ET mediated by tunnelling through a solvent barrier. We have performed further calculations to clarify this question further, and our results are presented below.

A clear way to distinguish between adiabatic and non-adiabatic ET is by evaluating the rate dependence on the electronic coupling, $V$, between the two ET states. A rate that follows Fermi's golden rule (and is therefore non-adiabatic) will increase with the coupling as $|V|^2$, whereas an adiabatic rate is expected to be weakly dependent on coupling up until $|V|$ is large enough to induce barrier-lowering effects. Although we don't have direct knowledge of the exact value of $V$ in our system, we can make inferences by computing both non-adiabatic and adiabatic rates at different couplings, and evaluating which theory is in best agreement with experimental data.

The model that we use for adiabatic ET differs slightly from the impurity model in Schmickler's formulation of the problem[78], which is the treatment adopted by Liu and colleagues[73]. We adopt an alternative, simpler model because we find it more amenable for a dielectric continuum description of the reorganization energy, and it allows us to implement the specific form of the DOS of MLG more easily[5]. In particular, we start with a $2 \times 2$ Hamiltonian describing the ET process between a discrete molecular state in the electrolyte and a specific electronic state $k$ in the electrode:

$$\mathbb{H}_k(\mathbf{q}) = \begin{pmatrix} H_{1,k}(\mathbf{q}) & V_k \\ V_k^* & H_{2,k}(\mathbf{q}) \end{pmatrix} \tag{33}$$

where $\mathbf{q}$ denotes nuclear coordinates. Going forward, we will assume a 'wide band approximation', meaning,

$$V_k = V = \text{const} \qquad \forall\, k \tag{34}$$

The classical non-adiabatic rate can be derived from this model by invoking Fermi's golden rule and linear response, resulting in the Marcus expression:

$$k_{1\to2}(\epsilon_k) = \frac{|V|^2}{\hbar} \sqrt{\frac{\pi}{k_B T \lambda}} \exp\left[ -\beta \frac{(\lambda + \Delta\epsilon - \epsilon_k)^2}{4\lambda} \right] \tag{35}$$

The electrochemical rate is then given by averaging over electronic states in the electrode giving the well-known result:

$$k_{1\to2} = \int D(\epsilon)(1 - f(\epsilon)) k_{1\to2}(\epsilon) d\epsilon \tag{36}$$

With electrode DOS $D(\epsilon)$, and Fermi–Dirac distribution, $f(\epsilon)$. An adiabatic rate can also be derived from the same model Hamiltonian in equation (33). We start by diagonalizing the $2 \times 2$ Hamiltonian, leading to an adiabatic Hamiltonian:

$$H_{\text{ad},k}(\mathbf{q}) = \frac{1}{2}(H_{1,k}(\mathbf{q}) + H_{2,k}(\mathbf{q})) - \frac{1}{2}\sqrt{(H_{1,k}(\mathbf{q}) - H_{2,k}(\mathbf{q}))^2 + 4|V|^2} \tag{37}$$

The adiabatic free energy surface associated with this Hamiltonian can be constructed from importance sampling in a molecular simulation. Alternatively, we can make the following simplifying assumption: within the linear response regime, we expect the adiabatic free energy surface to be given by a simple mixture of the corresponding diabatic (Marcus) free energy surfaces:

$$F_{\text{ad},k}(\Delta E) \approx \frac{1}{2}(F_{1,k}(\Delta E) + F_{2,k}(\Delta E)) - \frac{1}{2}\sqrt{\Delta E^2 + 4|V|^2} \tag{38}$$

where $\Delta E \equiv H_{2,k} - H_{1,k} = F_{2,k} - F_{1,k}$ is the vertical energy gap between the diabatic states, recognized in Marcus theory as the reaction coordinate, and $F_{1,2}$ have the usual parabolic form:

$$F_{1,k}(\Delta E) = \frac{(\Delta E + \lambda + \Delta\epsilon - \epsilon_k)^2}{4\lambda} \tag{39}$$

$$F_{2,k}(\Delta E) = \frac{(\Delta E - \lambda + \Delta\epsilon - \epsilon_k)^2}{4\lambda} + \Delta\epsilon - \epsilon_k \tag{40}$$

Equation (38) describes a bistable free energy surface, depicted in Extended Data Fig. 8, with shape defined by the reorganization energy, driving force and electronic coupling. The rate of transition between these meta-stable wells can be calculated using standard approaches such as transition state theory or Kramers' theory. The Kramers' estimate for the rate is:

$$k_{1\to2}^{\text{ad}}(\epsilon_k) = \frac{m\omega_1\omega_b}{2\pi\gamma} e^{-\beta\Delta F_{\text{ad}}^\ddagger} \tag{41}$$

where $\gamma$ is the solvent friction and:

$$\omega_1 = \sqrt{\frac{1}{m}\left(\frac{\partial^2 F_{\text{ad}}}{\partial \Delta E^2}\right)_{\Delta E = \Delta E_1}} \tag{42}$$

$$\omega_b = \sqrt{-\frac{1}{m}\left(\frac{\partial^2 F_{\text{ad}}}{\partial \Delta E^2}\right)_{\Delta E = \Delta E^\ddagger}} \tag{43}$$

$$\Delta F_{\text{ad}}^\ddagger = F_{\text{ad}}(\Delta E^\ddagger) - F_{\text{ad}}(\Delta E_1) \tag{44}$$

$\Delta E_1$ is the location of the reactant minimum, and $\Delta E^\ddagger$ is the location of the barrier. The dependence on the mass $m$ cancels out when inserting equations (42) and (43) into equation (41), and the dependence on the solvent friction cancels when evaluating ratios of rates. All of these quantities depend on reorganization energy, driving and coupling. The net adiabatic rate can finally be estimated by summing over all thermally accessible reactive channels in the electrode:

$$k_{1\to2}^{\text{ad}}(\lambda, V) = \int D(\epsilon)(1 - f(\epsilon)) k_{1\to2}^{\text{ad}}(\epsilon; \lambda, V) d\epsilon \tag{45}$$

$$= \frac{1}{2\pi\gamma} \int D(\epsilon)(1 - f(\epsilon)) \omega_1(\epsilon; \lambda, V) \omega_b(\epsilon; \lambda, V) e^{-\beta \Delta F_{\text{ad}}^\ddagger(\epsilon; \lambda, V)} d\epsilon \tag{46}$$

Note that every value of $\epsilon$ in the integrand defines a different $F_{\text{ad}}$, with different stationary points and frequencies that need to be calculated at every point when evaluating the integral through quadrature.

We applied this model of adiabatic ET to calculate rates in doped MLG. Calculations using equation (45) were performed using both our model for a screening-dependent reorganization energy and a constant value of reorganization energy, and then compared with the corresponding non-adiabatic rate. The results can be found in Extended Data Fig. 9. As expected, the adiabatic rate approaches the non-adiabatic limit as $|V| \to 0$, and the rate behaviour is even quite similar for a coupling of $5\,k_B T$. As the coupling increases to larger values, we begin to see that the rate enhancement is less pronounced and deviates significantly from experimental measurements. We also note that accounting for doping-dependent reorganization energy in the adiabatic rates is also crucial to improve agreement with the experimental rate enhancement, as we see in the non-adiabatic calculation.

In summary, these results strongly indicate that the coupling in our system is small enough to warrant a non-adiabatic treatment, and adiabatic rates with stronger coupling cannot explain our experimental results. These calculations, in conjunction with the aforementioned arguments in support of non-adiabaticity, allow us to confidently assume non-adiabatic behaviour.

## Data availability

Data supporting the findings of this study are available from the corresponding authors upon request. Source data are provided with this paper.

## Code availability

Code generated during the study is available from the corresponding authors on reasonable request.

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

**Acknowledgements** This material is based on work supported by the US Department of Energy, Office of Science, Office of Basic Energy Sciences, under award no. DE-SC0026007 (experimental studies by S.M., Y.W., M.P.E., J.N.N. and D.K.B.) and DEAC02-05-CH11231 in the Fundamentals of Semiconductor Nanowire Program (KCPY23) (theoretical work by L.C.E. and D.T.L.). Confocal Raman spectroscopy was supported by a Defense University Research Instrumentation Program grant through the Office of Naval Research under award no. N00014-20-1-2599 (D.K.B.). Fluorescence measurements by A.T. and K.X. were supported by STROBE, a National Science Foundation Science and Technology Center under grant no. DMR 1548924. K.W. and T.T. acknowledge support from the JSPS KAKENHI (grant nos. 21H05233 and 23H02052), the CREST (JPMJCR24A5), JST and World Premier International Research Center Initiative (WPI), MEXT, Japan.

**Author contributions** S.M. and D.K.B. conceived the study. S.M. and Y.W. fabricated the samples with assistance from M.P.E. J.N.N. grew the RuCl₃ crystals. S.M. and Y.W. performed the electrochemical measurements and COMSOL simulations. L.C.E. and D.T.L. performed the theoretical modelling. S.M. and M.P.E. performed the transport measurements. S.M. and A.T. performed the fluorescence measurements under the supervision of K.X. T.T. and K.W. provided the hBN crystals. S.M., L.C.E., D.T.L. and D.K.B. analysed the data. S.M., L.C.E., D.T.L. and D.K.B. wrote the paper.

**Competing interests** The authors declare no competing interests.

**Additional information**
**Correspondence and requests for materials** should be addressed to David T. Limmer or D. Kwabena Bediako.

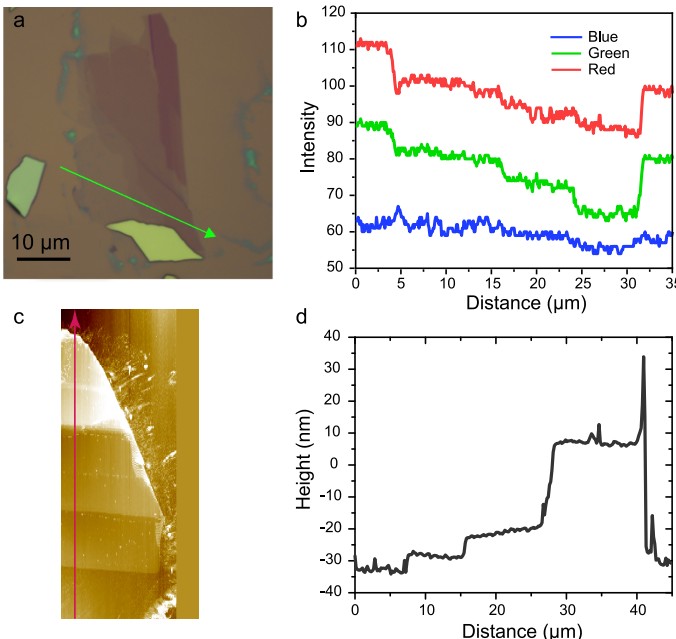

**Extended Data Fig. 1 | Optical and AFM characterization of exfoliated graphene and hBN.** (**a**) Optical image of a graphene flake comprising of monolayer, bilayer and trilayer graphene exfoliated on $SiO_2$/Si chips ($SiO_2$ thickness 285 nm). (**b**) Optical contrast (O.C.) line plot of red, green and blue channel intensities of panel **a** (along green arrow). (**c,d**) AFM line plot (along red arrow) of a multi-layer hBN exfoliated on $SiO_2$/Si chips ($SiO_2$ thickness 285 nm).

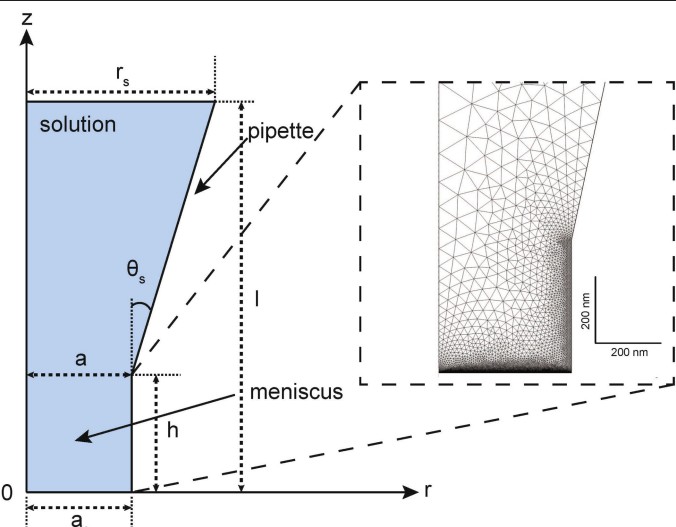

**Extended Data Fig. 2 | Model geometry for the COMSOL simulation.** Geometry of the simulation space and an example of mesh used for simulations (inset).

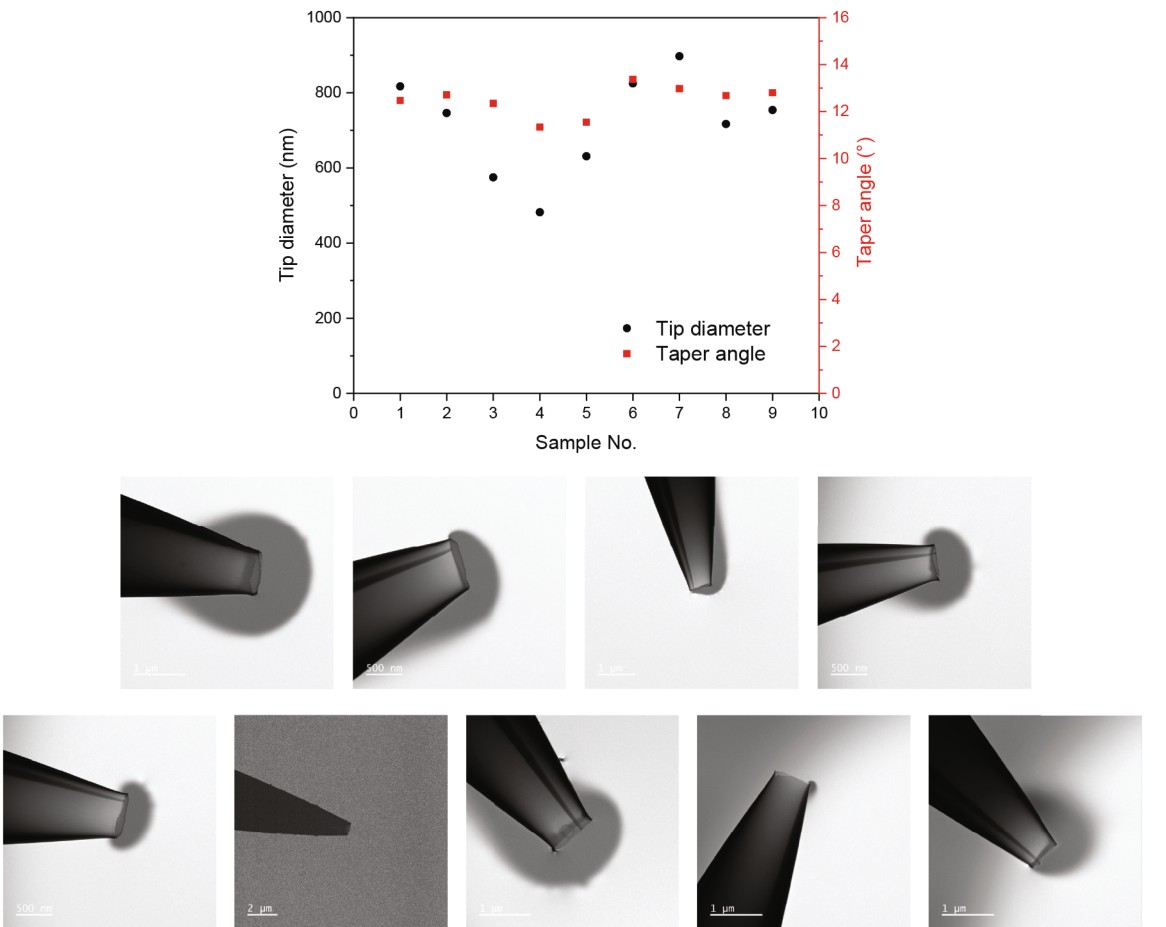

**Extended Data Fig. 3 | TEM bright field images of quartz pipettes. Top:** Survey of nanopipette geometries showing the taper angle and the orifice diameter distribution. **Bottom:** Representative TEM images of quartz pipettes used in SECCM.

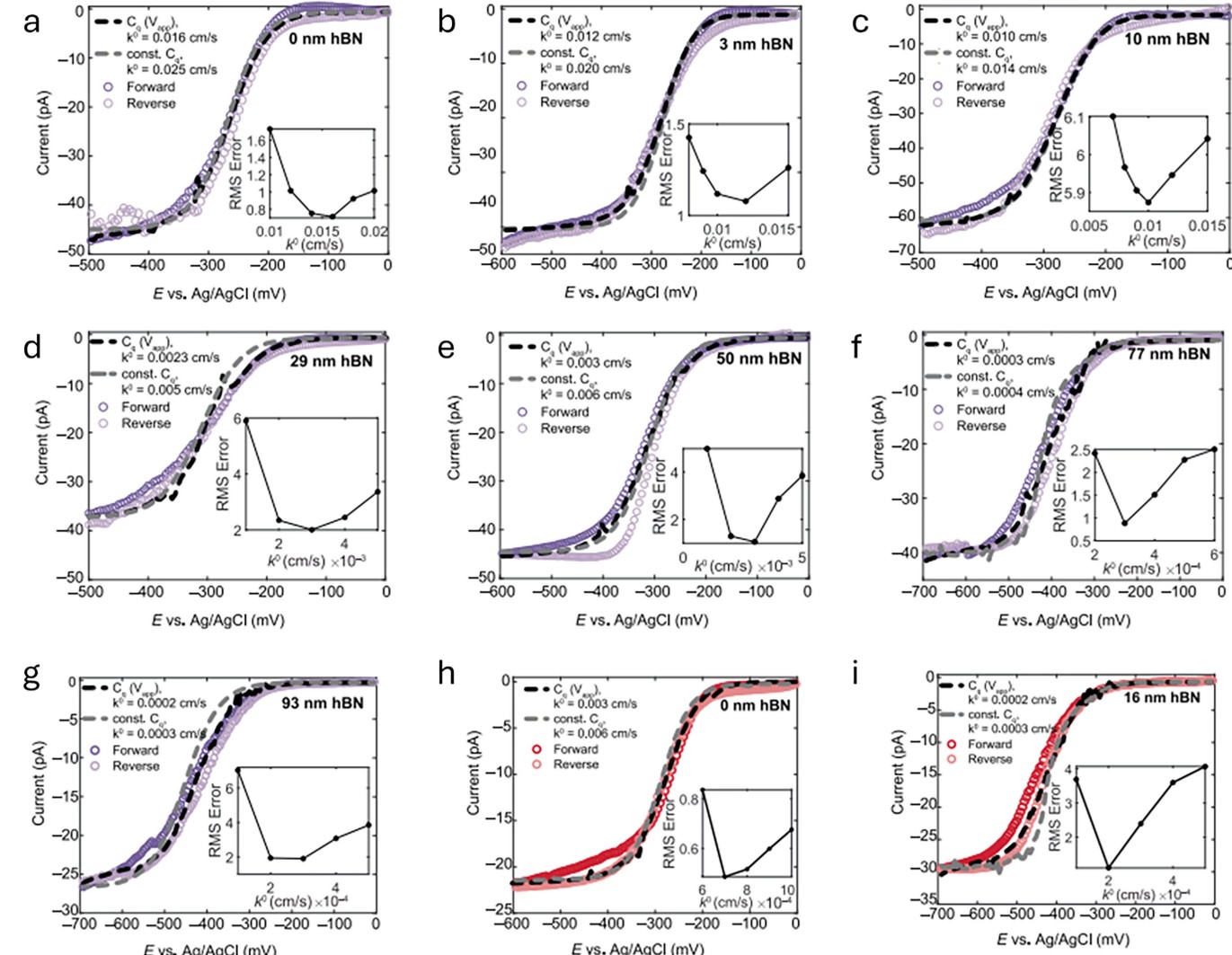

**Extended Data Fig. 4 | Representative experimental and simulated voltammograms. (a)–(i)** Cyclic voltammograms of $Ru^{3+/2+}$ on MLG/hBN/dopant electrodes (purple: $RuCl_3$ dopant, red: $WSe_2$ dopant) with associated COMSOL simulations assuming a potential-dependent $C_q$ (black dashed line) and constant $C_q$ (grey dashed line) at each hBN spacer thickness (Scan rate $v = 100$ mV/s). Inset: RMS error as a function of $k^0$ used in simulation.

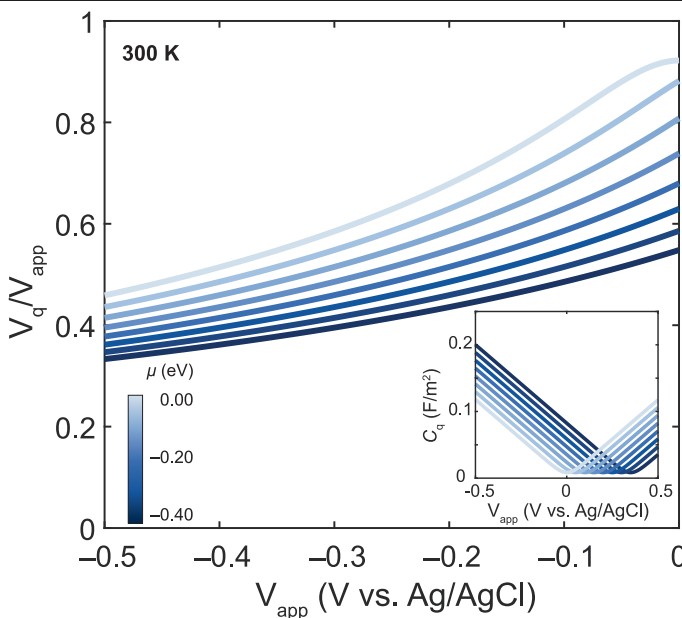

**Extended Data Fig. 5 | Potential-dependent quantum capacitance of graphene.** Ratio of $V_q/V_{app}$ as a function of applied potential. *Inset:* Quantum capacitance, $C_q/V_{app}$, versus applied potential.

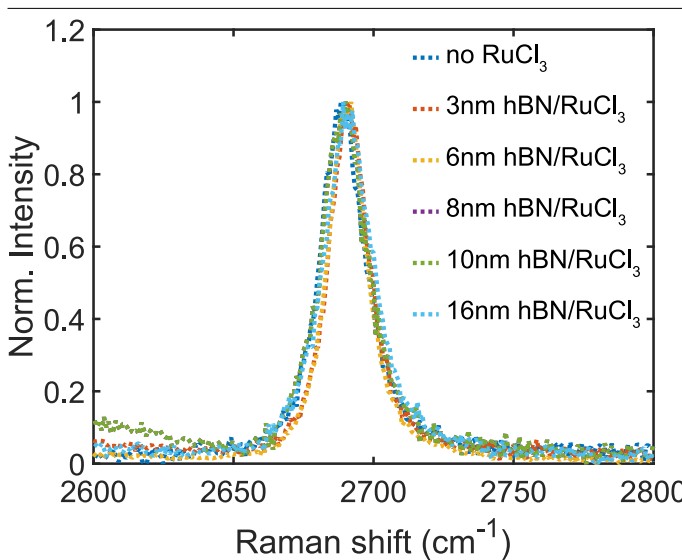

**Extended Data Fig. 6 | Raman spectra of the graphene 2D peak.** Negligible shifts in graphene 2D peak in MLG/hBN/RuCl$_3$ heterostructures with varying hBN thickness are consistent with minimal strain or geometric alteration of the graphene lattice.

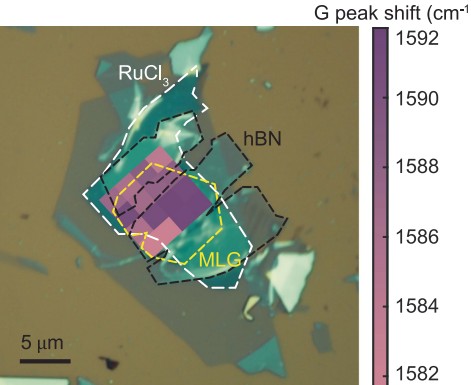

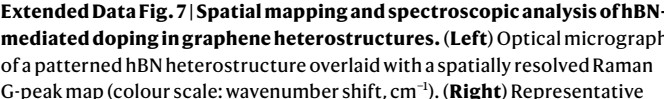

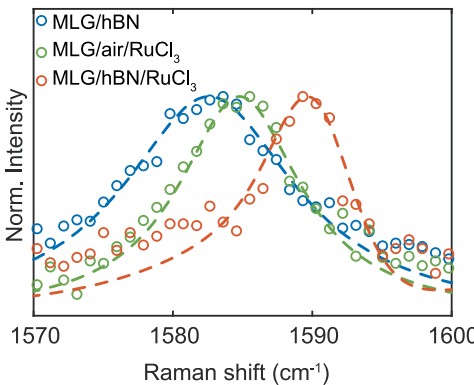

**Extended Data Fig. 7 | Spatial mapping and spectroscopic analysis of hBN-mediated doping in graphene heterostructures. (Left)** Optical micrograph of a patterned hBN heterostructure overlaid with a spatially resolved Raman G-peak map (colour scale: wavenumber shift, cm$^{-1}$). **(Right)** Representative G-peak spectra for MLG/hBN (blue), MLG/hBN/RuCl$_3$ (red), and MLG/air/RuCl$_3$ (green), with dashed lines indicating Voigt fits from which peak positions are obtained.

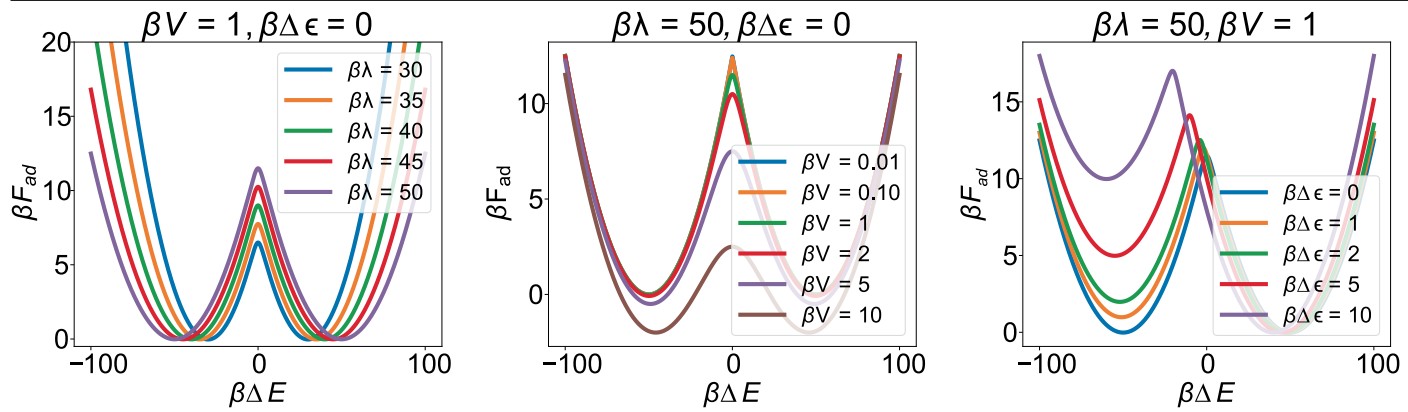

**Extended Data Fig. 8 | Adiabatic free energy projected along the energy gap coordinate. Left:** Varying $\lambda$, **Middle:** Varying $V$, **Right:** Varying $\Delta\epsilon$.

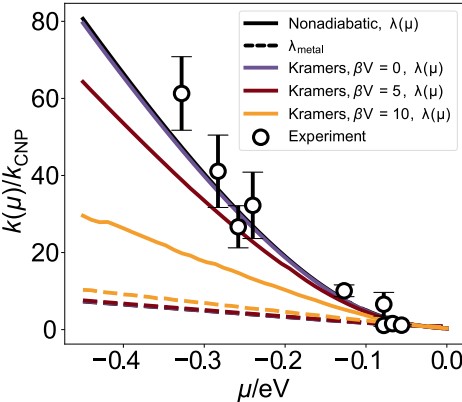

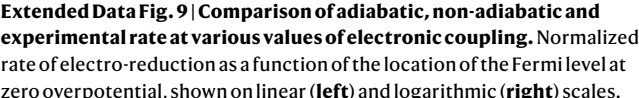

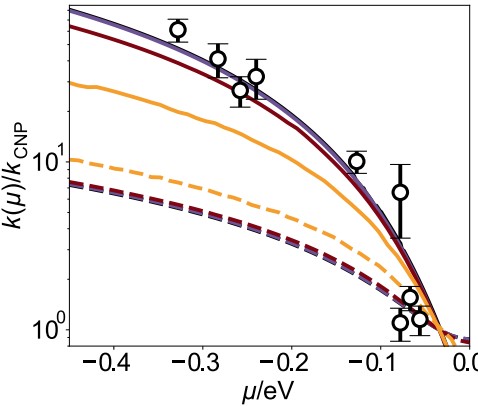

**Extended Data Fig. 9 | Comparison of adiabatic, non-adiabatic and experimental rate at various values of electronic coupling.** Normalized rate of electro-reduction as a function of the location of the Fermi level at zero overpotential, shown on linear (**left**) and logarithmic (**right**) scales.

Dashed lines correspond to the rate obtained from a constant reorganization energy, and solid lines doping-dependent reorganization energy. Error bars denote standard deviation for each experimental $k^0$ where n varies from 3–6.

**Extended Data Table 1 | $k^0$ (cm/s) as a function of hBN thickness for MLG regions with and without RuCl₃/WSe₂**

| hBN thickness (#cycles) | $k^0$ (MLG/RuCl$_3$ or *WSe$_2$) | $k^0$ (MLG) |
|---|---|---|
| 0(1) | 0.015, 0.016, 0.020 | n/a |
| 3(1) | 0.012, 0.01, 0.015, 0.012, 0.008 | n/a |
| 10(2) | 0.009, 0.009, 0.01, 0.015 | 0.0003, 0.0005, 0.0002 |
| 29(2) | 0.0022, 0.0023, 0.0023, 0.003 | 0.00022, 0.00026, 0.00025 |
| 50(2) | 0.002, 0.004, 0.004, 0.003, 0.001 | 0.0005, 0.0002, 0.0005, 0.0005 |
| 77(2) | 0.0004, 0.0004, 0.0003, 0.0003 | 0.00020, 0.00020, 0.00025, 0.00025 |
| 93(2) | 0.00020, 0.00020, 0.00025, 0.00030, 0.00030 | 0.00020, 0.00020, 0.00025 |
| 0*(1) | 0.007, 0.006, 0.007, 0.007, 0.010 | n/a |
| 16*(2) | 0.0001, 0.0002, 0.0002, 0.0002, 0.0002, 0.0002 | 0.00015, 0.00020, 0.00015 |

**Extended Data Table 2 | Comparison of calculated and experimental $E_f$ shifts in eV for $d$ = 3.89 nm**

| Configuration | Calculated | Experimental |
|---|---|---|
| Suspended | $-0.173 \pm 0.016$ | $-0.164$ |
| hBN-supported | $-0.250 \pm 0.021$ | $-0.303$ |