## [Peer Review File · Nature]

Electronic origin of reorganization energy in interfacial electron transfer

Corresponding Author: Professor Daniel Bediako

Version 0:

Reviewer comments:

Referee #1

(Remarks to the Author)

This is an outstanding and timely manuscript that makes a significant contribution to our understanding of electrode kinetics. The authors convincingly demonstrate the critical role of the electrode density of states (DOS) in modulating electron transfer (ET) rates. By leveraging atomically layered heterostructures, they are able to systematically tune the DOS of graphene and directly measure the resulting ET kinetics. These advances are made possible by recent developments in twistrionics and our growing understanding of graphene's electronic properties, allowing for a high degree of experimental control.

That said, the manuscript is quite dense and, in places, challenging to follow. I believe the clarity of the paper would benefit greatly from a few specific improvements and clarifications:

1. Reorganization Energy vs. Pre-exponential Factor:

The authors emphasize the effect of varying DOS on the reorganization energy. However, from a more intuitive perspective, one might expect the DOS to primarily influence the pre-exponential factor in ET kinetics. While I believe I understand the reasoning behind their focus, this conclusion may seem counterintuitive to many readers. I strongly encourage the authors to explain this point more clearly and accessibly—ideally in plain language—to help the broader community appreciate the underlying physics.

2. Relation between DOS and Thomas–Fermi Screening:

The manuscript makes several references to the Thomas–Fermi (TF) model and the effect of electric-field penetration into metals. The connection between DOS and TF screening is described as follows: They write:

"The screening length, ℓ_{TF} , is inversely proportional to the Fermi level, μ , reflecting enhanced metallicity as the charge carrier density increases. As $|\mu|$ increases (corresponding to an increased metallicity and decrease in ℓ_{TF}), the induced charge density becomes sharply localized, whereas a decrease in doping, or equivalently a decrease in metallicity, leads to a more diffuse charge distribution."

This is absolutely right. But I think the authors should dwell on the point, that they have this unique new option to vary at their will the TF-length, which was not possible in the earlier studies of ET on 'classical' metal electrodes. In this respect the authors do open a new page in ET science!

All in all, I suggest that the authors elaborate on this relationship and its relevance to their findings.

3. Earlier literature:

In this context, I would like to draw the authors' attention to an earlier work that seems particularly relevant. The effect of electric-field penetration (within the TF approximation) on reorganization and activation energies in ET was addressed in:

P.G. Dzhavakhidze et al., Activation Energy of Electrode Reactions: The Nonlocal Effects, *J. Electroanal. Chem.* 228 (1987) 329–346.

This work predicted a nontrivial, “anti-Marcusian” behavior—namely, that the reorganization energy increases as the reaction distance R to the electrode decreases, contrary to standard expectations. The physical origin of this effect was well explained in that paper. It would be very interesting to know whether the current authors’ model and calculations are consistent with that prediction.

Additional relevant insights were later discussed in:

D.K. Phelps et al., Nonlocal Electrostatic Effects on Electron-Transfer Activation Energies, *J. Phys. Chem.* 1990, 94, 1454–1463.

These earlier works predate the references currently cited by the authors (including the follow-up work by Medvedev), and I recommend that the authors consider citing and discussing them where appropriate.

4. Fig.3c is very interesting. Whereas the earlier, mentioned work of Dzhavakhidze et al studied the effect of R on reorganization energy, the authors study essentially the effect of varying TF-length, and there is some similarity in the rise of reorganization energy for low DOS. In any case the physics behind Fig.3c should be explained in more detail.

In summary, while these suggestions mainly pertain to clarity and context, the scientific content of the manuscript is highly compelling. With some careful editing and clarification, I believe this paper is fully deserving of publication in *Nature*.

Referee #2

(Remarks to the Author)

The authors use an innovative platform to address a fundamental question in electrochemical science: the effect of density of states on electrochemical kinetics. Even for simple classical outer sphere redox reactions (such as the one studied here), the role of density of states in electrochemical kinetics has generated debate for several decades. While this process is fairly easily studied at semi-conductor electrodes (and there is a long history), it has been much more difficult to discern effects on metal and carbon electrodes (notwithstanding some elegant work in the 1980s on molecularly -modified electrodes in the 1980s). More recent progress has been made by studying 2D materials and different numbers of layers of, for example, graphene, or by studying different numbers of layers of 2D materials on metals (e.g. graphene on as-grown copper) or transferred to other metals.

The corresponding author of this paper, and coworkers, introduced the idea of using twisted bilayer graphene to tune the DOS and investigate the effect on measured ET kinetics (*Nature Chem*; ref 18). Through this elegant approach, they demonstrated a strong effect, but also noted a major discrepancy between experiments and the classical Marcus-Hush-Chidsey theory. Here, using the same redox system, $[\text{Ru}(\text{NH}_3)_6(3+/2+)]$ they use solid state dopant layers and hexagonal boron nitride (hBN) spacers, to electrostatically tune the doping levels in monolayer graphene and obtain a detailed picture of how DOS affects the ET kinetics. As in their *Nature Chem* paper, they do not find agreement with the Marcus model, and - through modeling - suggest that this is due to a strong effect of the DOS on the reorganization energy, rather than just the number of thermally accessible channels. This is an important result that will generate much interest.

Superficially, the work builds on the earlier *Nature Chem* paper, but the new design of the 2D material and doping is actually a very clever way to be able to tease out the effects that are demonstrated.

The major conclusions of the paper rely on the integration of results from several key techniques. Some further details on the SECCM measurements and data are needed:

- The authors should be clear as to how many measurements were made for each hBN thickness (number of voltammograms and number of samples) and how the error bars in Fig 1d are arrived at.
- The rate constants cover a range of about 20 in magnitude and so the trend shown in Fig 1d seems reasonably convincing. However, I'd particularly ask the authors to look at the upper limit (graphite and thin hBN). The mass transport rate here is of the order of k_0 and so measuring k_0 becomes a bit more challenging. Likewise, the value for graphite is lower than for earlier measurements with SECCM on freshly cleaved graphite with the same couple $[\text{Ru}(\text{NH}_3)_6(3+/2+)]$, which found the reaction to be very close to the diffusion limit ($k_0 > 0.5 \text{ cm s}^{-1}$), but also found temporal effects (<https://doi.org/10.1002/anie.201200564>). The fast reaction and temporal effects was confirmed in further studies by macroscale measurements.
- The authors derive k_0 by using the B-V model (which is semi-empirical) and fitting to the experimental CV curves. One example fit is shown in Supplementary Fig 2 and the fit is not particularly good. A few points follow:
 - (a) I request that the authors include in Supplementary all the data (as raw I-V - not normalized) with the associated fits. I am guessing there are no more than 8 or 9 curves. The tips used were shown.
 - (b) As appropriate they should comment on the reasons for the non-goodness of fit (and also to clearly indicate the forward and back scans on the CVs). There are a few experimental factors at play here: e.g. at 100 mV s^{-1} , I suspect the response is not quite at true steady-state, yet the authors model it as such (eq 1), but the data could also be revealing of other factors.

(c) In Fig 1 (c) the authors could usefully include the tip sizes and limiting current values (or report the raw data in Supplementary). Each of these curves, with a different tip, will have a different mass transport rate (and being closer to or further from true steady-state).

(d) If possible, the authors should provide more robust evidence for the wetting, and the absence of electrowetting (measurements or prior literature).

- Following on, I wondered whether the authors could make more of the data, by making fuller use of the curves, for example by extracting potential-dependent effective rate constants? This could be informative since the DOS would be expected to vary significantly over the potential range, which is around the electroneutrality point for graphite, graphene etc. It would provide further confidence in the idea and the model. At the moment, all the data and analysis effectively relate to one potential (E_0 , k_0).

- For the general reader, it would also be useful for the authors to highlight why $[\text{Ru}(\text{NH}_3)_6(3+/2+)]$ is ideal for these measurements. It's been used quite a lot over the past decade or so and there is a rich literature.

Overall, the manuscript is readable and accessible. The idea is novel and interesting. I'd like to see the authors improve the presentation of the data (show all the raw electrochemical data - certainly more than one i-V curve), and to consider digging deeper into the data for the analysis. I think it would strengthen the argument.

I appreciate that the authors are writing a succinct piece, but they also missed quite a bit of literature on electrochemistry on 2D materials, including with $[\text{Ru}(\text{NH}_3)_6(3+/2+)]$, support effects, numbers of layers, history effects, etc.

Referee #3

(Remarks to the Author)
See attachment

Referee #4

(Remarks to the Author)

This report describes a detailed experimental and theoretical study of the electronic structure of the interface between (aqueous?) electrolyte and a composite graphene electrode bridged by variable-thickness (3-120 nm) hexagonal boron nitride (hBN) layers to (crystalline?) RuCl_3 or WSe_2 surfaces. RuCl_3 and WSe_2 are sources of hole and electron doping, respectively, hBN controlling the dopant level. The composite electrodes are characterized in detail by Raman spectroscopy and by Hall effect and fluorescence measurements, and used to map systematically the variation of the electrochemical rate constant of a standard $[\text{Ru}(\text{NH}_3)_6]^{2+/3+}$ probe redox couple with the hBN thickness and dopant level of the composite electrodes using Scanning electrochemical cell microscopy and cyclic voltammetry. The analysis is supported by "Finite element COMSOL Multiphysics simulations". The overarching outcome is the conclusion that nuclear reorganization not only on the solution side of the electrochemical interface but also on the electrode side contributes significantly to the activation (free) energy and rate constant of the electrochemical process. Overpotential dependent electronic contributions to electrochemical interfacial structures, electrocatalysis, and electrochemical processes have been addressed before. The results presented here could, however be of interest, but unfortunately the presentation and whole manuscript are fraught both with some general ("major") and some specific ("minor") shortcomings in the way of stringency and clarity, and in other ways:

Some general observations:

- "Defect-mediated charge transfer" makes good sense, and it is understandable that electron or hole transfer between the redox couple and localized electronic states in the electrode material can contribute to the reorganization energy. However, there is no allusion to the precise physical nature of the reorganization energy contribution from the electrode side, the identity of the nuclear modes reorganized, and the nature of the electronic-vibrational coupling. Neither is it clear, how the reorganization (free) energy was calculated, except for frugal allusions to notions such as "continuum model", "simulations"... I am afraid that this is overall very little satisfactory.

- The reference list to concepts and formalism of fundamental electron transfer theory is misleading. Most of the references listed in the first part of the manuscript are old with little allusion to the developments of electron transfer theory over the past 20+ years or longer. Neither is there any reference to the pioneering work by the Soviet and Russian school, Levich, Dogonadze, Kuznetsov and later by their associates. They could at least refer to the comprehensive review papers and books written by these authors in Western literature. Reference to Chidsey in this context is also misleading, as he simply used formalism developed by the Soviet school.

- Introduction of background and objectives of the study are only very frugally described, on p.4 lower part.

- I am aware that the theoretical model(s) used may be described in some detail in the Supplementary Material, but a brief overview should also be given in the main text.

Some specific observations:

- A rationale for the particular choice of multilayer system, graphene, hBN, RuCl_3 . WSe_2 is needed.

- I wonder if the Butler-Volmer limit is satisfactory, p. 6. The reorganization energy is usually reflected in non-linear effects

beyond this limit. This may be inherent in the procedures used, but is not clear and should be stated explicitly.

- I notice that the voltammetric shifts in Figure 1c amount to 25-50 mV or thereabout. This is only (1-2) kBT. Is this enough to be significant? And, why is there only a single point in the WSE2 curve in Figure 1d?
- The discussion on p. 9, both top and bottom is woolly and in need of significantly improved stringency. So is p. 10 top (“... derived from an attenuation.... How was it derived? = 0.82 eV for the redox pair alluded to on p. 9 bottom was not on pure gold but on gold modified by a self-assembled thiol molecular monolayer. And, why is the difference between the potential surfaces in the inset of Figure 3c so small, if the reorganization free energy contributions from the electrode are widely different?
- The authors have summarized their conclusions in the Discussion and Conclusion Section, p. 11. They could have taken the chance to address also here the precise physical origin of the reorganization features on the electrode side of the interface. The summary should clearly be worked over again.

In conclusion this work could be of interest as noted. However, at best massive re-writing, elaboration, clarification, and revision along the lines suggested is needed.

Version 1:

Reviewer comments:

Referee #1

(Remarks to the Author)

I am satisfied by the authors' response to my comments.

I had only very limited time to familiarize with the authors' response to three other referees, and am not in a position to comment on them, apart from one point, which looked off key to me: In appreciation of the pioneering contributions of Dogonadze's school, the authors cited a paper in Russian in proceedings of CITCE in conference in Moscow,

Levich, V. G. & Dogonadze, R. R. Osnovnie voprosi sovremenoi teoreticheskoi elektrokhemii in Proceedings of the 14th Meeting of the International Committee on Electrochemical Thermodynamics and Kinetics (CITCE) (Moscow, 1963), 21.

Who could ever read that? And it sounds, that there were no fundamental monographs that describe the problem in many aspects and contains a detailed review of the history of this dramatic field of research:

1. Electron Transfer in Chemistry and Biology: An Introduction to the Theory
Alexander M. Kuznetsov, Jens Ulstrup, ISBN: 978-0-471-96749-1; Wiley 1999, 376 pages
2. Charge Transfer in Physics, Chemistry and Biology: Physical Mechanisms of Elementary Processes and an Introduction to the Theory
A.M. Kuznetsov (1995) [eBook Published 23 September 2020] London, CRC Press, Pages 636.
DOI <https://doi.org/10.1201/9781003077244>

Citing those fundamental monographs would have restored the balance.

Referee #3

(Remarks to the Author)

I thank the authors for carefully addressing all the raised issues. The responses to my questions are very thorough and complete -- I am satisfied with the answers and don't have more questions.

That being said, I hope that the authors would consider augmenting the supporting information with the analyses and discussions on the adiabaticity vs non-adiabaticity and contrasting the current model with that of Liu et al.'s previous model. I found these discussions very interesting and helpful, and other readers might also find them insightful.

Finally, I wish to congratulate the authors on the exceptionally good work.

Referee #4

(Remarks to the Author)

The authors have responded meticulously to my suggestions to the first version of their manuscript, and as far as I can tell and as far as I can tell, also to those of the other reviewers. Most of my concerns are now relaxed, and the revised manuscript reads much more clearly and better than the first version. I have no wish to delay the further treatment of the manuscript, but to me a few – let us call them “minor” – points seem to remain. The authors could easily rectify these, should

they wish to do so.

I notice that the authors have now – meritoriously – inserted literature references to Kornyshev and associates and to Kuznetsov and associates. How the latter references are old, and there is no credit to Kuznetsov and associates' comprehensive work over the last several decades. There are also more accessible early references that the “#...osnovnie voprosi...” 1963 conference proceedings. This is not satisfactory and should be rectified, by reference also to Kuznetsov and associates' much more recent studies selected from books, review articles etc., or in other ways.

The discussion of the particular choice of $[\text{Ru}(\text{NH}_3)_6]^{2+/3+}$ as a prime probe redox couple is fine but short of a couple of points. The molecular structures of the oxidized and reduced forms of this couple are very similar, allowing to focus solely on solvent contributions to the reorganization free energy with intramolecular nuclear reorganization largely to be disregarded. Both forms are also robust with no risk of ligand substitution and other unwanted parasitic reactions such as for example for the $[\text{Co}(\text{NH}_3)_6]^{2+/3+}$ or $[\text{Fe}(\text{CN})_6]^{4-/3-}$ couples. This should be noted in the manuscript.

The discussion and answers to the reviewers regarding the adiabatic or diabatic behavior of the $[\text{Ru}(\text{NH}_3)_6]^{2+/3+}$ couple is broadly convincing, but reference to Important work by Nazmutdinov and associates (J. Phys. Chem. C 113 (2009) 2881-2890) could be given. This work analyzed the electronic coupling between mercury electrodes and $[\text{Ru}(\text{NH}_3)_6]^{2+/3+}$ along with other transition metal based redox probes. Although confined to metal electrodes, their results still offer important observations regarding differences between coupling via 3d and 4d orbitals of the central metal ion and concludes that the $[\text{Ru}(\text{NH}_3)_6]^{2+/3+}$ can be given the appellation of “weakly adiabatic”.

Without a wish of sounding too pedantic, the word “non-adiabatic” is a pleonasm (double negation). It should be replaced by “diabatic”. My conclusion is otherwise that the revised manuscript can be recommended for publication subject to attention to the points listed above.

Response to Reviewers' Comments

“Electronic origin of reorganization energy in interfacial electron transfer”

italics: Reviewers' comments

Blue: Our response

Red: Updated text

Referee 1

This is an outstanding and timely manuscript that makes a significant contribution to our understanding of electrode kinetics. The authors convincingly demonstrate the critical role of the electrode density of states (DOS) in modulating electron transfer (ET) rates. By leveraging atomically layered heterostructures, they are able to systematically tune the DOS of graphene and directly measure the resulting ET kinetics. These advances are made possible by recent developments in twistrionics and our growing understanding of graphene's electronic properties, allowing for a high degree of experimental control.

Response: We thank the reviewer for their positive and encouraging assessment. Our goal was to demonstrate how systematic control of the DOS can provide new opportunities for understanding and engineering electrode kinetics, and we greatly appreciate the recognition of this contribution.

That said, the manuscript is quite dense and, in places, challenging to follow. I believe the clarity of the paper would benefit greatly from a few specific improvements and clarifications:

1. Reorganization Energy vs. Pre-exponential Factor: The authors emphasize the effect of varying DOS on the reorganization energy. However, from a more intuitive perspective, one might expect the DOS to primarily influence the pre-exponential factor in ET kinetics. While I believe I understand the reasoning behind their focus, this conclusion may seem counterintuitive to many readers. I strongly encourage the authors to explain this point more clearly and accessibly—ideally in plain language—to help the broader community appreciate the underlying physics.

Response: We appreciate the opportunity to clarify this distinction and have made the following revisions to our conclusion:

“These results from experimental ET measurements and theoretical calculations demonstrate the pivotal influence of the electrode electronic structure on interfacial ET kinetics—**not only through the expected increase in the pre-exponential factor as DOS increases, but also through a significant impact on the reorganization energy itself.**”

2. Relation between DOS and Thomas–Fermi Screening:

The manuscript makes several references to the Thomas–Fermi (TF) model and the effect

of electric-field penetration into metals. The connection between DOS and TF screening is described as follows: They write:

“The screening length, ℓ_{TF} , is inversely proportional to the Fermi level, μ , reflecting enhanced metallicity as the charge carrier density increases. As $|\mu|$ increases (corresponding to an increased metallicity and decrease in ℓ_{TF}), the induced charge density becomes sharply localized, whereas a decrease in doping, or equivalently a decrease in metallicity, leads to a more diffuse charge distribution.”

This is absolutely right. But I think the authors should dwell on the point, that they have this unique new option to vary at their will the TF-length, which was not possible in the earlier studies of ET on ‘classical’ metal electrodes. In this respect the authors do open a new page in ET science! All in all, I suggest that the authors elaborate on this relationship and its relevance to their findings.

Response: We thank the reviewer for this insightful comment and agree that emphasizing the tunability of Thomas–Fermi screening length is essential to highlighting the novelty of our work. To clarify, we have added following lines to our introduction:

“A unique aspect of these electrode systems is the ability to continuously change the density of states at the Fermi energy, and correspondingly tune the ability of the electrode to screen electric fields. The Thomas-Fermi screening length, ℓ_{TF} , quantifies the length scale over which charges are screened in imperfect metals. Because ℓ_{TF} scales inversely with DOS, higher metallicity leads to sharper charge localization, whereas lower metallicity yields a more diffuse charge distribution. Such tunability offers a new avenue to investigate electronic screening shape interfacial ET.”

3. Earlier literature:

In this context, I would like to draw the authors’ attention to an earlier work that seems particularly relevant. The effect of electric-field penetration (within the TF approximation) on reorganization and activation energies in ET was addressed in:

P.G. Dzhavakhidze et al., *Activation Energy of Electrode Reactions: The Nonlocal Effects*, *J. Electroanal. Chem.* 228 (1987) 329–346.

This work predicted a nontrivial, “anti-Marcusian” behavior—namely, that the reorganization energy increases as the reaction distance R to the electrode decreases, contrary to standard expectations. The physical origin of this effect was well explained in that paper. It would be very interesting to know whether the current authors’ model and calculations are consistent with that prediction.

Additional relevant insights were later discussed in:

D.K. Phelps et al., *Nonlocal Electrostatic Effects on Electron-Transfer Activation Energies*, *J. Phys. Chem.* 1990, 94, 1454–1463.

These earlier works predate the references currently cited by the authors (including the follow-up work by Medvedev), and I recommend that the authors consider citing and discussing them

where appropriate.

Response: We appreciate the Reviewers’ suggested references, and we agree that they are extremely relevant for the present study. We are well-aware of the work done by Dzhavakhidze, Medvedev, Kornyshev, *et al*, and much of the theoretical treatment presented in this paper is founded on their pioneering contributions. As noted, we already cite some references from these authors. The 1990 J. Phys. Chem. publication is already being cited in our manuscript, and we have now included the earlier 1987 paper suggested by the reviewer.

As for the dependence of the reorganization energy on the separation from the electrode, R , our treatment is consistent with their original observations, but we focus our attention on a regime where the reorganization energy is still a monotonically increasing function of R . The interesting behavior that they report, where the reorganization energy decreases with increasing separation, comes about not only from electronic screening in the electrode, but also by introducing non-local electrostatic dispersion in the electrolyte, in the form of Debye-Hückel (ionic) screening. In our analysis, we model the electrolyte as a uniform dielectric, which corresponds to taking the Debye-Hückel screening length, Λ_{DH} , to zero. In Fig. 3 of the J. Electroanal. Chem. paper mentioned by the reviewer, it’s clear that the inverse relationship of the reorganization energy with R only arises for non-zero values of Λ_{DH} . We decided to neglect ionic screening effects mainly because we restrict our analysis to a fixed separation from the electrode, which we take as a parameter of the model. This means that, at a constant electrolyte concentration, the effect of ionic screening would just result in an additive factor to the reorganization energy that is independent of screening in the electrode. The trend observed in the rate as a function of doping would therefore not be altered by this effect at fixed R . The decision to model the electron transfer rate with a fixed separation from the electrode is justified by constant potential molecular dynamics simulations, which show that the double layer structure, inferred from the potential of mean force of a redox species as a function of R , is virtually insensitive to the degree of screening in the electrode. These results, shown in Fig. 1, imply that the thermal probability of a redox species to be at any separation R from the electrode surface, including the distance of closest approach, is to a good approximation independent of doping, justifying the use of a constant value of R for all the calculations.

That being said, we certainly agree that further studies aimed at reliably controlling the separation of the redox species from the electrode, and evaluating its effect on the rate in conjunction with other variables such as ionic concentration, would be very enlightening. They would help establish whether the prediction regarding ionic screening by Dzhavakhidze and co-workers is accurate, and it would shed light on the ‘optimal distance’ of electron transfer, which is often taken to be simply the distance of closest approach to the electrode.

4. *Fig.3c is very interesting. Whereas the earlier, mentioned work of Dzhavakhidze et al studied the effect of R on reorganization energy, the authors study essentially the effect of varying TF-length, and there is some similitaty in the rise of reorgnization energy for low DOS. In any case the physics behind Fig.3c should be explained in more detail.*

Figure 1: Potentials of mean force of Fe^{2+} (top) and Fe^{3+} (bottom) as a function of separation from the electrode surface, calculated at TF screening lengths of 0, 7.5 and 15 Å. The electrode surface is located at $z \approx 36$ Å, and the minimum for all screening lengths in this system is located at $z \approx 29$ Å. This indicates that the distance of closest approach of the redox species to the electrode is insensitive to screening length

In summary, while these suggestions mainly pertain to clarity and context, the scientific content of the manuscript is highly compelling. With some careful editing and clarification, I believe this paper is fully deserving of publication in Nature.

Response: Thank you for this encouraging feedback and the suggestion. In response, we have modified our explanation of the physics behind Fig. 3c in the revised manuscript. We appreciate the reviewer’s positive assessment and believe these clarifications will strengthen the manuscript’s clarity and broader relevance.

“Next, in Figure 3c, we account for this ℓ_{TF} to model a reorganization energy, $\lambda(\mu)$, that is a function of DOS and μ using non-local dielectric continuum theory. **Our model predicts a strong attenuation of $\lambda(\mu)$ with increasing metallicity (increasing DOS), converging to a value of 0.82 eV, consistent with λ_{metal}^1 . This trend arises from enhanced stabilization of the**

charge transfer transition state due to sharply localized polarization within the electrode. This manifests in the calculated free energy surfaces for the $\text{Ru}^{3+}/\text{Ru}^{2+}$ redox system at different values of μ (left inset of Figure 3c). These calculations reveal a pronounced influence of electrode screening on electron transfer kinetics: reduced μ monotonically increases the activation free energy (ΔG^\ddagger), corresponding to the intersection of the reactant and product free energy surfaces. The right inset of Figure 3c plots ℓ_{TF} vs. μ . The plot reveals a rapid decline in ℓ_{TF} as the system is doped away from the CNP ($\mu = 0$), directly linking doping-induced Fermi level shifts to screening efficiency. By explicitly including $\lambda(\mu)$ —which decreases with rising DOS—our model achieves quantitative agreement with experimental rates (red line in Figure 3a), establishing that electrode electronic structure governs λ and by extension the activation free energy landscape. Therefore, the predominant contributor to increasing ET rates with increasing DOS is the decrease in λ , not the change in number of thermally accessible electron donor/acceptor states.”

Referee 2

The authors use an innovative platform to address a fundamental question in electrochemical science: the effect of density of states on electrochemical kinetics. Even for simple classical outer sphere redox reactions (such as the one studied here), the role of density of states in electrochemical kinetics has generated debate for several decades. While this process is fairly easily studied at semiconductor electrodes (and there is a long history), it has been much more difficult to discern effects on metal and carbon electrodes (notwithstanding some elegant work in the 1980s on molecularly modified electrodes). More recent progress has been made by studying 2D materials and different numbers of layers of, for example, graphene, or by studying different numbers of layers of 2D materials on metals (e.g., graphene on as-grown copper) or transferred to other metals.

The corresponding author of this paper, and coworkers, introduced the idea of using twisted bilayer graphene to tune the DOS and investigate the effect on measured ET kinetics (Nature Chem., Ref. 18). Through this elegant approach, they demonstrated a strong effect, but also noted a major discrepancy between experiments and the classical Marcus–Hush–Chidsey theory. Here, using the same redox system, $[\text{Ru}(\text{NH}_3)_6]^{3+/2+}$, they use solid-state dopant layers and hexagonal boron nitride (hBN) spacers to electrostatically tune the doping levels in monolayer graphene and obtain a detailed picture of how DOS affects the ET kinetics. As in their Nature Chem. paper, they do not find agreement with the Marcus model and—through modeling—suggest that this is due to a strong effect of the DOS on the reorganization energy, rather than just the number of thermally accessible channels. This is an important result that will generate much interest.

Superficially, the work builds on the earlier Nature Chem. paper, but the new design of the 2D material and doping is actually a very clever way to be able to tease out the effects that are demonstrated.

The major conclusions of the paper rely on the integration of results from several key tech-

niques. Some further details on the SECCM measurements and data are needed:

Response: We thank the reviewer for their thorough and thoughtful evaluation of our work, and for recognizing both the novelty of our approach and the broader significance of our findings.

- The authors should be clear as to how many measurements were made for each hBN thickness (number of voltammograms and number of samples) and how the error bars in Fig 1d are arrived at.

Response: We thank the reviewer for this valuable point. Measurements were conducted on multiple independently fabricated devices, each featuring a distinct hBN thickness and comprising regions of evaporated Au as well as monolayer graphene with and without RuCl₃. Notably, the 0 and 3 nm hBN thickness data were measured on different regions the same device, as were the data for 77 and 93 nm hBN. For devices without hBN, RuCl₃ and WSe₂ are sensitive to air exposure, so the entire monolayer graphene was used to encapsulate them, and consequently, no isolated MLG regions were available.

For each device, we typically recorded 1–2 voltammetric cycles at multiple spatially separated positions to ensure reproducibility and capture local variability. Voltammetric data from each MLG position, including multiple cycles, were binned and individually fitted to COMSOL simulations to extract k^0 values. The Au regions served as an internal reference, with their CVs fit using a reversible rate constant of 0.5 cm/s to determine the correct E^0 for each sample and measurement. This procedure yielded multiple k^0 values per device. The values plotted in Fig. 1d represent averages across these measurements, with error bars indicating the standard deviation. All extracted k^0 values are provided in Table 1.

Table 1: **(Methods, Table 1)** k^0 (cm/s) as a function of hBN thickness for MLG regions with and without RuCl₃/WSe₂.

hBN thickness, nm		
(# cycles)	k^0 (MLG/RuCl ₃ or *WSe ₂)	k^0 (MLG)
0 (1)	0.015, 0.016, 0.020	n/a
3 (1)	0.012, 0.01, 0.015, 0.012, 0.008	n/a
10 (2)	0.009, 0.009, 0.01, 0.015	0.0003, 0.0005, 0.0002
29 (2)	0.0022, 0.0023, 0.0023, 0.003	0.00022, 0.00026, 0.00025
50 (2)	0.002, 0.004, 0.004, 0.003, 0.001	0.0005, 0.0002, 0.0005, 0.0005
77 (2)	0.0004, 0.0004, 0.0003, 0.0003	0.00020, 0.00020, 0.00025, 0.00025
93 (2)	0.00020, 0.00020, 0.00025, 0.00030, 0.00030	0.00020, 0.00020, 0.00025
0* (1)	0.007, 0.006, 0.007, 0.007, 0.010	n/a
16* (2)	0.0001, 0.0002, 0.0002, 0.0002, 0.0002, 0.0002	0.00015, 0.00020, 0.00015

These methodological details and table (Table 1) have been added to **Section 1.2 SECCM measurements** in the Methods (page 4: para 2, 3).

- *The rate constants cover a range of about 20 in magnitude and so the trend shown in Fig 1d seems reasonably convincing. However, I'd particularly ask the authors to look at the upper limit (graphite and thin hBN). The mass transport rate here is of the order of k^0 and so measuring k^0 becomes a bit more challenging. Likewise, the value for graphite is lower than for earlier measurements with SECCM on freshly cleaved graphite with the same couple $[Ru(NH_3)_6^{3+/2+}]$, which found the reaction to be very close to the diffusion limit ($k^0 > 0.5 \text{ cm s}^{-1}$), but also found temporal effects (<https://doi.org/10.1002/anie.201200564>). The fast reaction and temporal effects was confirmed in further studies by macroscale measurements.*

Response: Thank you for highlighting these important considerations. We agree that when the electron transfer rate approaches the mass transport–limited regime, precise determination of k^0 becomes inherently more challenging due to the convolution of kinetic and transport effects. However, our observed rates for doped monolayer graphene are $\leq 0.02 \text{ cm/s}$, and for graphite approximately 0.03 cm/s , indicating that electron transfer remains primarily kinetically controlled within the experimental window.

Our reported graphite values are lower than reported in some previous studies, which can be attributed to differences in surface preparation, measurement delay, and environmental conditions. Notably, the referenced study also reports that exposing the basal graphite surface to air for less than an hour after cleavage causes a measurable decrease in electron transfer rates. Although our devices were measured on the same day as fabrication, a 4–5 hour interval was required for device assembly—including stacking, making electrical contacts, and transfer to the SECCM measurement substrate—which may have contributed to slightly lower observed rates. In this context, having monolayer graphene regions without $RuCl_3/WSe_2$ on the same device provides a robust baseline to reliably study the relative enhancement in electron transfer kinetics induced by doping.

To clarify this for our readers, following details have been added in the Methods:

Section 1.3 Finite-element simulations

...The diffusion coefficients D_O and D_R for the $Ru(NH_3)_6^{3+/2+}$ couple were set to $8.43 \times 10^{-6} \text{ cm}^2/\text{s}$ and $1.19 \times 10^{-5} \text{ cm}^2/\text{s}$, respectively. $\alpha = 0.5$ was used for all simulations, consistent with previous studies on graphene thin films. **Our observed rates for doped monolayer graphene are $\leq 0.02 \text{ cm/s}$, and for graphite approximately 0.03 cm/s , indicating that electron transfer remains primarily kinetically controlled within the experimental window.**

Section 1.2 SECCM measurements (last para)

Although our devices were measured on the same day as fabrication, a 4–5 hour interval was required for device assembly—including stacking, making electrical contacts, and transfer to the SECCM measurement substrate—which may have contributed to the slower observed rates than reported in literature. In this context, having monolayer graphene regions without $RuCl_3/WSe_2$ on the same device provides a robust baseline to reliably study the relative

enhancement in electron transfer kinetics induced by these dopants.

- The authors derive k^0 by using the B-V model (which is semi-empirical) and fitting to the experimental CV curves. One example fit is shown in Supplementary Fig 2 and the fit is not particularly good. A few points follow:

(a) I request that the authors include in Supplementary all the data (as raw I-V - not normalized) with the associated fits. I am guessing there are no more than 8 or 9 curves. The tips used were shown.

Response: We thank the reviewer for this constructive suggestion. In total, we examined nine different hBN thicknesses, and for each thickness several k^0 values were determined, both with and without RuCl₃ treatment. This results in more than sixty individual I-V plots, which we feel would overwhelm the Supplementary Information and obscure the key trends. Instead, in the revised manuscript we have included representative I-V data and corresponding fits. We have included one exemplar voltammogram for each hBN thickness 2 and two corresponding best fits—one which considers a constant quantum capacitance (C_q) value and the other which fits the CV using a C_q that varies with the applied potential. These data are now shown in **Supplementary Figure 4**, which is also shown here on page 9 below for convenience). In addition, we are now also providing *all* extracted k^0 values for every device and thickness, together with the relevant CV metadata, as Supplementary Data in Excel file format.

The following details have been added to the manuscript:

1. Representative I-V data and corresponding fits for each hBN thickness as Supplementary Figure 4
2. All binned and simulation data used for error plots to extract k^0 values have been uploaded as source data.

(b) As appropriate they should comment on the reasons for the non-goodness of fit (and also to clearly indicate the forward and back scans on the CVs). There are a few experimental factors at play here: e.g. at 100 mV s⁻¹, I suspect the response is not quite at true steady-state, yet the authors model it as such (eq 1), but the data could also be revealing of other factors.

Response: Thank you for the insightful question. The fit presented was generated by binning multiple cycles of forward and backward scans and calculating the RMS deviation relative to simulated COMSOL data. Consequently, the fit represents an averaged behavior over many cycles, and therefore, slight deviations between individual experimental cycles and the modeled fit are expected. We now also show nine other experimental CVs (both forward and backward scans) and fits in Figure 2 below (**Supplementary Figure 4** in the revised manuscript), and we consider these fits to be very reasonable, especially when the potential-dependence of quantum capacitance effects is taken into account (Black dashed lines).

Figure 2: **(Methods, Supplementary Fig. 4) Representative experimental and simulated voltammograms.** Cyclic voltammograms of $\text{Ru}^{3+/2+}$ on MLG/hBN/dopant electrodes (purple: RuCl_3 dopant, red: WSe_2 dopant) with associated COMSOL simulations assuming a potential-dependent C_q (black dashed line) and constant C_q (grey dashed line) at each hBN spacer thickness (Scan rate $v = 100$ mV/s). *Inset*: RMS error as a function of k^0 used in simulation.

We also note that as doping of monolayer graphene increases, asymmetries in the DOS accessible during reduction and oxidation (on either side of the Fermi level) should lead to subtle but systematic differences between the forward and backward scans. This also contributes to imperfect fits. Additionally, as the reviewer noted, the assumption of true steady-state conditions—particularly at the scan rate of 100 mV s^{-1} —is an approximation. Non-steady-state phenomena such as finite diffusion layers or quasi-reversible kinetics might also manifest as small deviations from the modeled curves. Despite these complexities, the overall best fit curves remain very good across most extracted k^0 values.

Most importantly, the magnitude of variation in k^0 associated with RuCl_3 doping far exceeds the effects of any of these comparatively minor uncertainties in fitting, affirming the reliability of our central conclusions.

(c) In Fig 1 (c) the authors could usefully include the tip sizes and limiting current values (or report the raw data in Supplementary). Each of these curves, with a different tip, will have a different mass transport rate (and being closer to or further from true steady-state).

Response: We thank the reviewer for this helpful suggestion. We have now provided the raw I–V curves along with the corresponding tip sizes in the Supplementary Information. We also note that for the 77 nm and 93 nm RuCl_3 devices, the Au and graphite control data are identical, as they were measured on the same device; this clarification has been added to the corresponding excel sheet. We have also provided all binned and simulated data used to derive the error plots and extract k^0 values.

Corresponding Excel file: Raw data_CVsforK0.xlsx

Error plots folder: "Error calculations data".

(d) If possible, the authors should provide more robust evidence for the wetting, and the absence of electrowetting (measurements or prior literature).

Response: We appreciate reviewer’s insightful comment regarding wetting and electrowetting effects. Prior literature indicates that graphene and related layered materials generally exhibit stable wetting, with minimal evidence of electrowetting under typical electrochemical conditions. For instance, contact angle studies report consistent wettability of graphene, with only a modest decrease from 105° at 0 s to 90° after 4 minutes, significantly longer than the timescale of our measurements ($< 10 \text{ s}$).² Molecular dynamics simulations further suggest that increasing surface charge reduces wettability, implying that electrowetting effects should be even less pronounced in our doped devices.³ In addition, electrowetting measurements on HOPG in 0.1 M KF (potential range 0 to -2.0 V vs Ag/AgCl) demonstrate minimal electrowetting, consistent with our experimental conditions (0.1 M KCl, 0 to -0.7 V vs Ag/AgCl)⁴. Experimentally, the stability of our cyclic voltammetry signals and post-measurement microscopy further confirm the absence of significant electrowetting-induced morphological changes. Taken together, these results support that electrowetting does not influence our electrochemical measurements. Following discussion and references have been added to the manuscript:

Methods, section 1.2 SECCM measurements: “Prior contact angle studies on graphene report modest changes (from 105° to 90°) over several minutes,² which is significantly longer

than our measurement timescale (< 30 s). Molecular dynamics simulations show that increasing surface charge reduces wettability,³ suggesting that electrowetting effects should be even weaker in doped graphene. Consistently, electrowetting experiments on HOPG in 0.1 M KF over a potential window of 0 to -2.0 V vs. Ag/AgCl revealed negligible effects,⁴ in agreement with our experimental conditions (0.1 M KCl, 0 to -0.7 V vs. Ag/AgCl). Our cyclic voltammetry signals remained stable, and microscopy before and after testing confirmed no detectable morphological changes. These observations indicate that electrowetting does not significantly affect our measurements.”

- Following on, I wondered whether the authors could make more of the data, by making fuller use of the curves, for example by extracting potential-dependent effective rate constants? This could be informative since the DOS would be expected to vary significantly over the potential range, which is around the electroneutrality point for graphite, graphene etc. It would provide further confidence in the idea and the model. At the moment, all the data and analysis effectively relate to one potential (E_0 , k_0).

Response: We thank the reviewer for highlighting this important consideration. We agree that a systematic treatment of potential-dependent effective rate constants provides a clearer perspective on the role of DOS modulation. In our original analysis, we assumed constant values of quantum capacitance ($C_q = 0.1$ F/m²) and double-layer capacitance ($C_{dl} = 0.1$ F/m²). This was motivated by the fact that varying C_q between 0.05 and 0.15 F/m² while keeping C_{dl} constant produced only modest changes in the extracted k^0 , far smaller than the order-of-magnitude enhancements observed experimentally. This behavior is shown in Figure 3 below, where simulated CVs for $k^0 = 0.01$, 0.001, and 0.0001 with $C_q = 0.05$, 0.10, and 0.15 F/m² reveal only subtle differences.

Figure 3: **Simulated cyclic voltammograms** for representative k^0 values (0.01, 0.001, and 0.0001 cm/s) at different quantum capacitances ($C_q = 0.05$, 0.10, and 0.15 F/m²), assuming constant $C_{dl} = 0.1$ F/m².

However, we recognize that for a more rigorous treatment, it is important to account for these

Figure 4: **(Methods, Supplementary Fig. 5) Potential-dependent quantum capacitance of graphene.** Ratio of V_q/V_{app} as a function of applied potential. *Inset:* Quantum capacitance, C_q/V_{app} , versus applied potential.

effects systematically, and therefore, we have now incorporated potential-dependent C_q into our modeling framework. To do this, we reanalyzed all our data by incorporating a variable C_q as a function of V_{app} . These results (Figure 4 below and **Supplementary Fig. 5** in the revised manuscript) presents both $C_q(V_{\text{app}})$ and the associated $V_q(V_{\text{app}})$, which are now integrated into our COMSOL simulations (shown in Figure 3 above, which is **Supplementary Figure 4** in the revised manuscript. This updated approach yields potential-dependent effective rate constants and enables a more rigorous comparison to the expected DOS variation.

Notably, the revised rate constants (Figure 5) show even stronger dependence of rate constant with DOS and improved agreement with our theoretical predictions using the continuum model for $\lambda(\mu)$. **Figures 1** and **Figure 3** in the main text have now been updated to reflect these updated k^0 values, and Figure 4 above is now also provided in the Supplementary Information.

The following details have been added to the **Methods/Section 1.4 Quantum Capacitance:**

For monolayer graphene, the quantum capacitance C_q can be expressed as

$$C_q = e^2 \frac{dn}{dV_q}, \quad (1)$$

where e is the elementary charge, and $\frac{dn}{dV_q}$ is the rate of change of carrier concentration n with respect to V_q . Under the two-dimensional free-electron gas model, considering graphene's

Figure 5: **Left: (Main text, Fig. 1d)** Updated k^0 as a function of hBN thickness. Error bars indicate the standard deviation, **Right: (Main text, Fig. 3a)** Updated k^0 as a function of μ , normalized to the rate constant of undoped graphene, $k^{0'}$.

linear DOS near the Dirac point⁵, this relation simplifies to

$$C_q = \frac{2e^2 k_B T}{\pi \hbar^2 v_F^2} \ln \left[2 \left(1 + \cosh \left(\frac{eV_{\text{ch}}}{k_B T} \right) \right) \right], \quad (2)$$

where \hbar is the reduced Planck constant, k_B is the Boltzmann constant, $v_F \approx c/300$ is the Fermi velocity of Dirac electrons, and $V_{\text{ch}} = E_F/e$ is the graphene potential. At the Dirac point, where carrier concentration n is minimal, C_q approaches zero.

At $T = 300$ K, the channel potential can be written as

$$eV_{\text{ch}} = \mu + eV_{\text{app}}. \quad (3)$$

Assuming constant charge, the relationship between C_q and double-layer capacitance C_{dl} is

$$\frac{C_q}{C_{dl}} = \frac{V_{dl}}{V_q}. \quad (4)$$

Substituting $C_{dl} = 0.1$ F/m² gives

$$V_{dl} = 10 C_q V_q. \quad (5)$$

This leads to the relation between V_q and the applied potential V_{app} :

$$V_{\text{app}} = (1 + 10 C_q) V_q. \quad (6)$$

Finally, substituting Eq. 2 into Eq. 6 yields

$$\frac{V_q}{V_{\text{app}}} = 1 + 10 \cdot \frac{2e^2 k_B T}{\pi \hbar^2 v_F^2} \ln \left[2 \left(1 + \cosh \left(\frac{eV_{\text{ch}}}{k_B T} \right) \right) \right]. \quad (7)$$

This expression provides V_q/V_{app} as a function of V_{app} , from which $V_{dl}(V_{app})$ is extracted for different values of μ and incorporated into our COMSOL simulations to systematically account for quantum capacitance effects. Figure 4 shows the ratio V_q/V_{app} as a function of applied potential at 300 K. The *inset* presents the corresponding quantum capacitance, C_q , as a function of V_{app} .

- For the general reader, it would also be useful for the authors to highlight why $[Ru(NH_3)_6]^{3+/2+}$ is ideal for these measurements. It's been used quite a lot over the past decade or so and there is a rich literature.

Response: We thank the reviewer for this valuable suggestion. The $[Ru(NH_3)_6]^{3+/2+}$ redox couple is particularly well suited for our measurements because of its well-characterized, reversible, outer-sphere electron transfer behavior with no detectable adsorption on graphite electrodes as validated by *in situ* Raman spectroscopy.⁶⁻⁸ This ensures sensitivity to the electronic properties of the electrode while avoiding complications from surface-specific reactions. These details have also been emphasized and supported with relevant literature in the **Methods, Section: 1.2 SECCM measurements, para 2:**

$[Ru(NH_3)_6]^{3+/2+}$ was chosen as the redox couple because it exhibits well-characterized, reversible, outer-sphere electron transfer with no detectable adsorption on graphite electrodes, as confirmed by *in situ* Raman spectroscopy.⁶⁻⁸ This ensures that the measured kinetics are sensitive to the electronic properties of the electrode while avoiding complications from surface-specific reactions.

Overall, the manuscript is readable and accessible. The idea is novel and interesting. I'd like to see the authors improve the presentation of the data (show all the raw electrochemical data - certainly more than one i-V curve), and to consider digging deeper into the data for the analysis. I think it would strengthen the argument.

Response: We thank the reviewer for their encouraging comments and valuable suggestions. To improve data presentation and transparency, we have provided raw data underlying all plots in the Supplementary data. We have also updated the Methods section to include the complete set of calculated k^0 for all hBN thicknesses, along with the number of CV cycles and additional analysis details. Representative binned I-V data and corresponding fits, along with all error calculations and simulated plots, have likewise been included in the Supplementary Information. Furthermore, we have reanalysed our data incorporating the quantum capacitance effect, recognizing that DOS vary over the applied potential range. Collectively, these improvements enhance the rigor and clarity of our study and strengthen the support for the conclusions presented.

I appreciate that the authors are writing a succinct piece, but they also missed quite a bit of literature on electrochemistry on 2D materials, including with $[Ru(NH_3)_6]^{3+/2+}$, support effects, numbers of layers, history effects, etc.

Response: We thank the reviewer for this important feedback. We fully agree that there is a rich body of literature on the electrochemistry of 2D materials that provides valuable

context to our work. In revising this manuscript, we are mindful of the editor’s request for brevity and could not provide a comprehensive account here. To ensure that interested readers have access to this broader perspective, we have cited a recent review article that we co-authored with collaborators at George Mason University, where these studies are surveyed in detail and discussed in relation to emerging trends in 2D electrochemistry.[9]

Referee 3

Referee report on the manuscript "Electronic origin of the reorganization energy in interfacial electron transfer" by Maroo et al on June 28, 2025.

By using state-of-the-art electrochemical experiments, the authors show notable changes in the outer-sphere electron transfer (OS-ET) kinetics as the substrate underlying a graphene sheet is modulated. In particular, it is found that changing the thickness of a hexagonal boron nitride (hBN) spacer between graphene and RuCl₃ offers a way to control the OS-ET kinetics. An original theoretical model is developed to explain these results in a conceptually novel way and the authors propose that the hBN spacer thickness, i.e. the electrode, controls the OS-ET kinetics by changing the reorganization energy, which is a key parameter in ET theory and which is usually assumed to depend only on the electrolyte properties; to my knowledge, this is the first work to propose that rather subtle changes to the electrode can strongly influence the reorganization energy and thereby the OS-ET kinetics. This is a notable finding because the conventional wisdom in electrochemistry considers that OS-ET kinetics are usually rather independent of the electrode material (see some references in the comments). While recent studies have challenged this conventional view, especially on graphene electrodes (again, see some references below), variations in the OS-ET kinetics as well as electrode-driven changes in the reorganization energy reported in this work are exceptionally high.

Overall, I consider that this work challenges our current understanding of OS-ET theory and kinetics. Because OS-ET reactions and theories are the corner-stone of all electrochemical reactions, the fundamental understanding gained in the present work may substantially influence both the theory of electrochemical kinetics and the development of novel electrode materials for various electrochemical technologies. The main findings and theoretical model should, however, be more critically compared with previous studies and theories on OS-ET kinetics on graphene-based electrodes. This is needed to place the present work in the broader context of OS-ET reactions on graphene materials and to provide further proof that the reorganization energy is indeed controlled by the electrode potential by eliminating other possibilities.

Response: We are grateful to the reviewer for the close reading of our work, and recognizing its value and novelty. We have seriously considered all of the queries and requests for clarification put forward, please see below for our responses.

Major issues

1. *The authors present that the observed changes in OS-ET kinetics are due to changes in*

the reorganization energy resulting from modulations in the electrode electronic structure and the image potential localization in the electrode due to the substrate underlying the graphene surface. Given that the image charge interaction is electro-static in nature, the observed changes in the electron transfer kinetics appear to be due electrostatic effects. A similar conclusion has been reached for OS-ET on copper-supported graphene electrodes, for which it was shown that changes in the kinetics are also due to variations in the electrostatic potential (<https://www.nature.com/articles/s41467-021-27339-9>). However, this previous work attributes the electro-static potential changes to changes in the surface charge rather than the electrode metallicity or the reorganization energy. This long prelude leads to a question and a request:

(a) Further demonstrations of the electrode-controlled reorganization energy is needed. For this it is necessary to consider the option that the changes in OS-ET kinetics could be due to changes in the surface charge rather than the electrode-driven reorganization energy. To resolve whether changes in the electrostatics and OS-ET kinetics are due to changes in the surface charge as proposed previously or the electrode metallicity and the reorganization as proposed herein, I ask the authors to consider additional experiments (e.g. capacitance measurements and/or Kelvin probe force microscopy) to study whether the surface charge varies as a function of the hBN thickness.

(b) I ask the authors to carefully compare the findings, methods, and conclusions of the present and previous works (see also references in following points) – I believe this is needed to obtain a deeper perspective and understanding on OS-ET kinetics on 2D electrodes.

Response: We find the Reviewer’s suggestion to contrast our findings with those of Liu and co-workers¹⁰ to be both pertinent and helpful to clarify a very crucial point. The Reviewer is correct when pointing out that the changes in the reorganization energy that we observe as a function of doping are mediated by electrostatic interactions. However, that is where similarities between our insights and those explained as surface charge effects by Liu et al.¹⁰ end, and there is a very important distinction that again emphasizes the key fundamental insight of this work.

While our observation emphasizes how image charge interactions change with doping, we find that surface charge effects alone cannot explain the exponential enhancement in the observed electron transfer rate. Rather, it is the impact that these electrostatic interactions have on the reorganization energy that ultimately accounts for this novel phenomenon. We have undertaken molecular dynamics simulations, discussed in the following paragraphs, to further show that although net charge density can slightly alter the reorganization energy, the spatial distribution of this charge, set by screening and metallicity, exerts a much stronger influence on the OS-ET kinetics than the total surface charge. We now explain this in detail.

Firstly, we would like to point out that a material’s ability to accumulate (surface) charge and its metallicity are two intrinsically related properties. Under constant potential conditions, the charge on the electrode is determined by the material’s capacitance, which in turn depends on metallicity. Within the Thomas-Fermi model, this relationship is explicit: the screening length controlling metallicity is inversely related to the surface charge density. For

Figure 6: Reorganization energy under constant potential (of 0 V) and constant charge conditions. Q_{elec} denotes the net value of charge imposed on the electrode, and ρ is the corresponding charge density. Changing ℓ_{TF} at constant charge modifies the spatial distribution of the charge.

graphene, it takes the form:

$$\ell_{\text{TF}} = \frac{1}{2\pi e^2 D(E_F)} = \frac{v_F}{4\sqrt{\pi|\rho|}} \quad (8)$$

where $D(E_F)$ is the DOS evaluated at the Fermi energy, v_F is the Fermi velocity, and ρ is the surface charge density. Increasing $D(E_F)$ through doping simultaneously enhances metallicity (reduces screening) and increases the electrode’s charge density and polarizability, thereby implicitly accounting for its ability to accumulate charge.

While higher metallicity enhances polarization within the electrode, the converse is not true: a larger net surface charge alone does not substantially influence electron transfer. Moreover, the spatial distribution of surface charge, rather than its average magnitude, plays a dominant role in determining the reorganization energy, and this distribution is strongly governed by the doping level. Under constant charge conditions, higher metallicity produces a more sharply localized polarization response upon electron transfer, which impacts the electron transfer rate more significantly than the net charge itself.

To illustrate this, we performed constant potential and constant charge molecular dynamics simulations while varying the metallicity and calculated approximate reorganization energies, shown in Fig. 6. We find that introducing a net charge on the electrode has minimal impact on the reorganization energy at low screening, and even at stronger screening, the change due to charging is much smaller (a few $k_B T$) than the variation induced by changing the metallicity ($\sim 30 k_B T$). This demonstrates that, although net charge density can slightly modify the reorganization energy, the spatial distribution of this charge—determined by screening and metallicity—has a far greater effect on the OS-ET kinetics.

2. *Related to the previous point, the authors say that the hBN-modulated carrier density shifts the (electro)chemical potential of electrons of the graphene layer. I find this misleading*

as the electrochemical potential should be to constant across the system at a given electrode potential (on the absolute potential scale, the electrode potential is equal to the Fermi level as well as the electrochemical potential of electrons).

Because the electrochemical potential of electrons for a given phase i depends on the standard part ($\mu_{0,i}$), the electron activity ($a_{ie}(\mathbf{r})$), and the electrostatic inner potential $\phi_i(\mathbf{r})$ as $\mu_e = \mu_{0,i} - \phi_i(\mathbf{r}) + k_B T \ln(a_{ie}(\mathbf{r}))$, I would rather expect that changing the carrier concentration/density in hBN would influence the electrostatic potential and/or charge distribution within the system. These can, e.g., change the filling of the graphene DOS and its surface charge, which would in turn influence the electron transfer kinetics and/or the metallicity of graphene, as discussed in point 1. DFT calculations might help to resolve this issue.

Response: The reviewer’s comment highlights an important nuance, and we concur that the electrochemical potential remains constant at equilibrium across the electrode system. In our work, the introduction of solid-state dopants such as RuCl₃ and WSe₂ modulates the electronic band structure of graphene by stabilizing or destabilizing energy levels, effectively shifting bands relative to the electrochemical potential—or, equivalently, shifting μ **relative to its band structure**. This modulation alters the local electrostatic potential and charge distribution within the graphene, which has been extensively characterized using ARPES, Raman spectroscopy, Hall measurements, and ab initio theoretical approaches such as MINT and DFT^{11–13}. We have revised the manuscript to clarify these points for the readers at following places:

Main text, Figure 2 (d): “Schematic illustration of band alignment and interfacial charge transfer between graphene and α -RuCl₃, depicting E_F shifts **relative to its band structure...**”

Effect of electrode metallicity on the reorganization energy, para 1 “By relating the hBN-modulated carrier densities (Figure 2c) to shifts in the chemical potential, μ , (**i.e., E_F at equilibrium**) of MLG **relative to its band structure...**”

3. In the abstract it is written that “This work challenges a traditional paradigm of heterogeneous ET kinetics and establishes a fundamental role of the electrode electronic structure in interfacial reactivity.” I find this misleading: the electrode electronic structure is known to play a key role in interfacial electrochemistry as practically all inner-sphere electron transfer reactions and electrocatalytic reactions depend on the electronic structure. It would be more correct to say that outer-sphere electron transfer kinetics are usually assumed to be independent of the electronic structure (<https://doi.org/10.1016/j.coelec.2019.11.003>, <https://pubs.acs.org/doi/abs/10.1021/acs.chemrev.1c00583>); however, also many outer-sphere reaction are known to depend on the electronic structure (e.g. <https://doi.org/10.1016/j.coelec.2024.101632>)

Response: We appreciate the reviewer’s insightful clarification. We agree that the electrode electronic structure’s importance is well established not only in inner-sphere and electrocatalytic reactions but also in OSET reactions. In fact, in many cases, the absence of modulation of k^0 with DOS results from the DOS remaining sufficiently high, as in metals and graphite, masking any observable variation. Our aim, however, is to highlight that conventional models attribute DOS effects solely to the number of thermally accessible electronic states, as

embodied in conventional formulations. These models typically neglect electrode polarization effects during electron transfer, which introduce an additional energy penalty via the reorganization energy λ , particularly in systems with limited DOS. While λ has traditionally been treated as independent of electrode electronic structure, our findings challenge this assumption by demonstrating a significant influence of electronic structure. We have made the following revisions to the abstract to better reflect this perspective:

“...We find the ensuing variation in ET rate arises from strong modulation in a reorganization energy associated with image potential localization in the electrode. **This work redefines the traditional paradigm of heterogeneous ET kinetics, revealing a deeper role of the electrode electronic structure in interfacial reactivity.**”

4. *The authors write that the introduction of RuCl₃ or WSe₂ influence the electrode through the donation of electrons/holes and do not e.g. change the electrode geometry. Is there some proof for this?*

Response: We thank the reviewer for this query. Raman spectroscopy of monolayer graphene provides a sensitive probe of strain and morphological changes, particularly via the 2D peak position. To address this, we have included a new figure (Figure 7 below and **Supplementary Figure 6** in the revised manuscript) showing negligible shifts in the 2D peak for monolayer graphene with and without hBN/RuCl₃, across varying hBN thicknesses, confirming the absence of significant geometric alteration. Moreover, the use of charge-transfer materials to modulate doping in van der Waals systems is known to tune electronic properties without affecting surface morphology. This methodology is extensively supported in the literature, where complementary spectroscopic and microscopic techniques have consistently demonstrated electronic modification in the absence of structural change.¹⁴⁻¹⁶ We have added this plot and following details to the Methods section for clarity:

Figure 7: **(Methods, Supplementary Fig. 6)** Raman spectra of the graphene 2D peak for MLG/hBN/RuCl₃ heterostructures with varying hBN thickness.

1.5.2 Spectra acquisition and analysis: “...A constant background is subtracted from each spectrum before fitting the peaks. The 2D peak position provides a sensitive probe of strain and morphological changes. Supplementary Figure 4 shows negligible shifts in the 2D peak for monolayer graphene with and without hBN/RuCl₃, across varying hBN thicknesses, thereby confirming the absence of significant geometric alteration.”

5. Figure 2d provides a schematic illustration of the band alignment across RuCl₃/hBN/MLG systems. In the text, this alignment is proposed to result in enhanced ET kinetics due to the downward shift in the Fermi-level and the related increase in the availability of electronic states for ET. The schematic and qualitative picture could be improved and made more quantitative by carrying out DFT simulations of the RuCl₃/hBN/graphene electrode surfaces to study the electrostatic potential and DOS profiles across the system. By further relating the Fermi level with the absolute redox potential of [Ru(NH₃)₆]^{3+/2+} (see <https://www.nature.com/articles/s41524-023-01184-4>, Figure 5), the authors can strengthen their view on the importance of the higher DOS in increasing OS-ET kinetics.

Response: We thank the reviewer for this valuable suggestion. The doping modulation of graphene and associated band alignment using RuCl₃ and WSe₂ has been extensively characterized through ARPES, Raman spectroscopy, Hall measurements, and ab initio theoretical approaches, including MINT and DFT.^{11–14,16,17} In addition, previous work from our group investigated this heterostructure in the context of Li intercalation, where DFT band-structure calculations were performed; these results are shown below in Fig.8. We also show results from other studies on such systems (Figs.9, 10) and have cited the relevant literature in our manuscript.

Figure 8: Band structure and DOS for a MLG/ α -RuCl₃ heterostructure. MLG-derived bands and DOS are shown in blue, and α -RuCl₃-derived bands and DOS are in orange. E_F = Fermi level. MLG charge-neutrality point is indicated by dashed gray line.¹⁷

Figure 9: Band alignment schematic; the work function difference between α - RuCl_3 and other compounds yields charge transfer¹¹

Figure 10: (a) Band alignment in the WSe_2 -graphene heterostructure. (b) Normalized PL spectra of WSe_2 samples with various thicknesses.¹⁶

6. The OS-ET kinetics are modeled using an expression valid for non-adiabatic ET reactions (Eq. 15-16). However, previous work has shown that OS-ET on graphene electrodes for the redox couple considered in this work ($[\text{Ru}(\text{NH}_3)_6]^{3+}/2+$) is adiabatic (<https://www.nature.com/articles/s41467-021-27339-9>). Also, the small DOS at the graphene electrode does not necessarily imply that the ET would be non-adiabatic (<https://doi.org/10.1016/j.electacta.2022.140901>). While the impact of the coupling constant V in Eq.15 will likely cancel out when comparing the ET kinetics across different $\text{RuCl}_3/\text{hBN}/\text{graphene}$ systems and therefore not impact the main conclusion, is there some evidence for assuming that the considered ET reaction is non-adiabatic?

Response: We thank the referee for raising this important point. The question of non-adiabatic vs. adiabatic outer sphere electron transfer in electrochemistry is a contentious topic that is often hard to settle definitively. We appreciate the Reviewers' inquiry, as it prompted us to consider this matter more carefully. We have strong reasons to believe

that electron transfer in our system proceeds non-adiabatically. On top of experimental evidence in this and other works in support of this perspective, we have carried out additional numerical tests confirming that we can safely exclude being within the adiabatic regime.

Firstly, we find that the rate is highly sensitive to the electronic properties of the electrode, and this is conventionally regarded as a distinctive characteristic of non-adiabatic electron transfer.

Furthermore, there is contrasting experimental evidence to that presented in¹⁰ in regard to the adiabaticity of OS-ET in the $[\text{Ru}(\text{NH}_3)_6]^{3+/2+}$ redox couple at graphene electrodes. The alluded work posits adiabatic electron transfer based on the assumption that increasing graphene layers equates to increasing tunneling distance. However, this perspective may neglect the role of graphene’s intrinsic electronic states, which actively participate in electron transfer, making the simplified tunneling argument inadequate, as we have illustrated in this work. Conversely, studies employing hBN as a true inert spacer on graphite electrodes indeed demonstrate an exponential decrease in electron transfer rates with increasing spacer thickness, consistent with non-adiabatic tunneling processes^{18,19}. The pronounced dependence of electron transfer kinetics on the electrode’s DOS observed here and in past studies further supports this interpretation²⁰⁻²².

Finally, electron transfer for the $[\text{Ru}(\text{NH}_3)_6]^{3+/2+}$ system has been reported to occur near the outer Helmholtz plane, where the redox species is separated from the electrode surface by a structured solvent layer. Experimental evidence, including negative activation volumes, indicates electron transfer through solvated species rather than direct electrode contact, implying weak electronic coupling²³. Collectively, these observations align with our assumption of non-adiabatic electron transfer mediated by tunneling through a solvent barrier. Nonetheless we have carried out additional calculations to clarify this question further, and our results are presented below.

A clear way to distinguish between adiabatic and non-adiabatic electron transfer, is by evaluating the rate dependence on the electronic coupling, V , between the two electron transfer states. A rate that follows Fermi’s Golden Rule (and is therefore non-adiabatic) will increase with the coupling as $|V|^2$, whereas an adiabatic rate is expected to be weakly dependent on coupling up until $|V|$ is large enough to induce barrier-lowering effects. While we don’t have direct knowledge of the exact value of V in our system, we can make inferences by computing both non-adiabatic and adiabatic rates at different couplings, and evaluating which theory is in best agreement with experimental data.

The model that we use for adiabatic electron transfer differs slightly from the impurity model in Schmickler’s formulation of the problem²⁴, which is the treatment adopted by Liu *et al* in the Nature Communications paper cited by the Reviewer¹⁰. We adopt an alternative, simpler model because we find it more amenable for a dielectric continuum description of the reorganization energy, and it allows us to implement the specific form of the density of states of monolayer graphene more easily²⁵. In particular, we start with a 2×2 Hamiltonian describing the electron transfer process between a discrete molecular state in the electrolyte

and a specific electronic state k in the electrode:

$$\mathbb{H}_k(\mathbf{q}) = \begin{pmatrix} H_{1,k}(\mathbf{q}) & V_k \\ V_k^* & H_{2,k}(\mathbf{q}) \end{pmatrix} \quad (9)$$

Where \mathbf{q} denotes nuclear coordinates. Going forward, we will assume a ‘wide band approximation’, meaning,

$$V_k = V = \text{const} \quad \forall k \quad (10)$$

The classical non-adiabatic rate can be derived from this model by invoking Fermi’s Golden Rule and linear response, resulting in the Marcus expression:

$$k_{1 \rightarrow 2}(\epsilon_k) = \frac{|V|^2}{\hbar} \sqrt{\frac{\pi}{k_B T \lambda}} \exp \left[-\beta \frac{(\lambda + \Delta\epsilon - \epsilon_k)^2}{4\lambda} \right] \quad (11)$$

The electrochemical rate is then given by averaging over electronic states in the electrode giving the well-known result:

$$k_{1 \rightarrow 2} = \int D(\epsilon)(1 - f(\epsilon))k_{1 \rightarrow 2}(\epsilon)d\epsilon \quad (12)$$

With electrode density of states $D(\epsilon)$, and Fermi-Dirac distribution, $f(\epsilon)$.

An adiabatic rate can also be derived from the same model Hamiltonian in Eq. 9. We start by diagonalizing the 2×2 Hamiltonian, leading to an adiabatic Hamiltonian:

$$H_{\text{ad},k}(\mathbf{q}) = \frac{1}{2} (H_{1,k}(\mathbf{q}) + H_{2,k}(\mathbf{q})) - \frac{1}{2} \sqrt{(H_{1,k}(\mathbf{q}) - H_{2,k}(\mathbf{q}))^2 + 4|V|^2} \quad (13)$$

Figure 11: Adiabatic free energy projected along the energy gap coordinate. **Left:** Varying λ , **Middle:** Varying V , **Right:** Varying $\Delta\epsilon$

The adiabatic free energy surface associated with this Hamiltonian can be constructed from importance sampling in a molecular simulation. Alternatively, we can make the following simplifying assumption: within the linear response regime, we expect the adiabatic free

energy surface to be given by a simple mixture of the corresponding diabatic (Marcus) free energy surfaces:

$$F_{\text{ad},k}(\Delta E) \approx \frac{1}{2} (F_{1,k}(\Delta E) + F_{2,k}(\Delta E)) - \frac{1}{2} \sqrt{\Delta E^2 + 4|V|^2} \quad (14)$$

where $\Delta E \equiv H_{2,k} - H_{1,k} = F_{2,k} - F_{1,k}$ is the vertical energy gap between the diabatic states, recognized in Marcus theory as the reaction coordinate, and $F_{1,2}$ have the usual parabolic form:

$$F_{1,k}(\Delta E) = \frac{(\Delta E + \lambda + \Delta\epsilon - \epsilon_k)^2}{4\lambda} \quad (15)$$

$$F_{2,k}(\Delta E) = \frac{(\Delta E - \lambda + \Delta\epsilon - \epsilon_k)^2}{4\lambda} + \Delta\epsilon - \epsilon_k \quad (16)$$

Equation 14 describes a bistable free energy surface, depicted in Fig. 11, with shape defined by the reorganization energy, driving force and electronic coupling. The rate of transition between these meta-stable wells can be calculated using standard approaches such as transition state theory or Kramers' theory. The Kramers' estimate for the rate is:

$$k_{1 \rightarrow 2}^{\text{ad}}(\epsilon_k) = \frac{m\omega_1\omega_b}{2\pi\gamma} e^{-\beta\Delta F_{\text{ad}}^\ddagger} \quad (17)$$

where γ is the solvent friction and:

$$\omega_1 = \sqrt{\frac{1}{m} \left(\frac{\partial^2 F_{\text{ad}}}{\partial \Delta E^2} \right)_{\Delta E = \Delta E_1}} \quad (18)$$

$$\omega_b = \sqrt{-\frac{1}{m} \left(\frac{\partial^2 F_{\text{ad}}}{\partial \Delta E^2} \right)_{\Delta E = \Delta E^\ddagger}} \quad (19)$$

$$\Delta F_{\text{ad}}^\ddagger = F_{\text{ad}}(\Delta E^\ddagger) - F_{\text{ad}}(\Delta E_1) \quad (20)$$

ΔE_1 is the location of the reactant minimum, and ΔE^\ddagger is the location of the barrier. The dependence on the mass m cancels out when inserting Eqs. 18 and 19 into 17, and the dependence on the solvent friction cancels when evaluating ratios of rates. All of these quantities depend on reorganization energy, driving and coupling. The net adiabatic rate can finally be estimated by summing over all thermally-accessible reactive channels in the electrode:

$$k_{1 \rightarrow 2}^{\text{ad}}(\lambda, V) = \int D(\epsilon)(1 - f(\epsilon))k_{1 \rightarrow 2}^{\text{ad}}(\epsilon; \lambda, V)d\epsilon \quad (21)$$

$$= \frac{1}{2\pi\gamma} \int D(\epsilon)(1 - f(\epsilon))\omega_1(\epsilon; \lambda, V)\omega_b(\epsilon; \lambda, V)e^{-\beta\Delta F_{\text{ad}}^\ddagger(\epsilon; \lambda, V)}d\epsilon \quad (22)$$

Note that every value of ϵ in the integrand defines a different F_{ad} , with different stationary points and frequencies that need to be calculated at every point when evaluating the integral through quadrature.

Figure 12: Normalized rate of electro-reduction as a function of the location of the Fermi level, at zero overpotential. Comparison of adiabatic, non-adiabatic and experimental results at various values of electronic coupling. Dashed lines corresponds to the rate obtained from a constant reorganization energy, solid lines doping-dependent reorganization energy.

We applied this model of adiabatic electron transfer to calculate rates in doped monolayer graphene. Calculations using Eq. 21 were performed using both our model for a screening-dependent reorganization energy and a constant value of reorganization energy, and subsequently compared to the corresponding non-adiabatic rate. The results can be found in Fig. 12. As expected, the adiabatic rate approaches the non-adiabatic limit as $|V| \rightarrow 0$, and even for a coupling of $5 k_B T$, the rate behavior is quite similar. As the coupling increases to larger values, we begin to see that the rate enhancement is less pronounced and deviates significantly from experimental measurements. Additionally, we note that accounting for doping-dependent reorganization energy in the adiabatic rates is also crucial to improve agreement with the experimental rate enhancement, as we see in the non-adiabatic calculation.

In summary, these results strongly indicate that the coupling in our system is small enough to warrant a non-adiabatic treatment, and adiabatic rates with stronger coupling cannot explain our experimental results. These calculations, in conjunction with the aforementioned arguments in support of non-adiabaticity, allow us to confidently assume non-adiabatic behavior.

Other issues

- *In my opinion, Chidsey's contribution to electron transfer theory cannot be considered seminal as the inclusion of the Fermi-Dirac distribution and the DOS was done well before Chidsey by Levich, Dogonadze, and Kuznetsov et al (e.g. [https://doi.org/10.1016/0079-6816\(75\)90008-8](https://doi.org/10.1016/0079-6816(75)90008-8)). That said, the application of the Marcus-Hush-Chidsey formula by Chid-*

sey in Ref.15 is indeed seminal.

Response: We appreciate the reviewer’s observation regarding foundational electron transfer literature. We agree that original manuscript did not fully acknowledge the seminal contributions from the Soviet and Russian schools of electrochemistry, especially the quantum mechanical approaches of Levich, Dogonadze, Chizmadzhev, Christov, and Kuznetsov. To address this, we have added the following text to the introduction to highlight their seminal contributions: The quantum mechanical theory of ET, pioneered by Levich, Dogonadze, Chizmadzhev, Christov, and Kuznetsov^{26,27}, similarly leads to Marcus–Hush-type rate expressions.

We have also updated our manuscript to include following articles from these foundational works:

1. Dzhavakhidze, P. G., Kornyshev, A. A. & Krishtalik, L. I. Activation energy of electrode reactions: the non-local effects. *J. Electroanal. Chem. Interfacial Electrochem.* **228**, 329–346 (1987).
2. Phelps, D. K., Kornyshev, A. A. & Weaver, M. J. Nonlocal electrostatic effects on electron-transfer activation energies: some consequences for and comparisons with electrochemical and homogeneous-phase kinetics. *J. Phys. Chem.* **94**, 1753–1761 (1990).
3. Levich, V. G. & Dogonadze, R. R. *Osnovnie voprosi sovremenoj teoreticheskoj elektrokhemii*. In *Proc. 14th Mtg. Int. Comm. Electrochem. Thermodyn. Kinet.* 21 (CITCE, Moscow, 1963).
4. Dogonadze, R. R., Kuznetsov, A. M. & Chernenko, A. A. Theory of homogeneous and heterogeneous electronic processes in liquids. *Russ. Chem. Rev.* **34**, 759–775 (1965).

• *In general, the reorganization energy is not limited to the solvent nuclei or solvent molecules, and the reorganization energy can depend also on the reorganization of the electrode surface.*

Response: We interpret this comment as suggesting that, beyond solvent reorganization, lattice vibrations and other electrode dynamics may also contribute to the reorganization energy in ET reactions. This is an interesting observation, and we agree that such effects are indeed relevant for inner-sphere ET, where the redox species adsorbs or bonds directly to the electrode; in these cases, lattice motions or bond-length changes can contribute appreciably to λ , particularly under strong coupling.^{28,29} By contrast, in OSET—such as the $[\text{Ru}(\text{NH}_3)_6]^{3+/2+}$ system studied here—electron transfer proceeds without direct bonding or significant lattice perturbation. Previous studies have shown that this redox couple remains solvent-separated from the electrode²³, and under such conditions, lattice vibrational coupling does not contribute measurably to λ .

• *μ is used on page 8 but it is first defined on page 10 as the Fermi level of the system. Furthermore, it would be helpful to be more explicit and call μ as the electrochemical potential of the electrons as commented on in point 2.*

Response: We thank the reviewer for pointing this out. We have corrected the text so that μ is introduced at its first occurrence (p. 8) and explicitly defined as the electrochemical potential of the electrons, which is equivalent to the Fermi level of the system at equilibrium. We believe following clarification addresses the concern and improves readability for the broader audience:

Effect of electrode metallicity on the reorganization energy, para 1: "By relating the hBN-modulated carrier densities (Figure 2c) to shifts in the chemical potential, μ , (i.e., E_F at equilibrium) of MLG relative to its band structure..."

Referee 4

This report describes a detailed experimental and theoretical study of the electronic structure of the interface between (aqueous?) electrolyte and a composite graphene electrode bridged by variable-thickness (3-120 nm) hexagonal boron nitride (hBN) layers to (crystalline?) RuCl₃ or WSe₂ surfaces. RuCl₃ and WSe₂ are sources of hole and electron doping, respectively, hBN controlling the dopant level. The composite electrodes are characterized in detail by Raman spectroscopy and by Hall effect and fluorescence measurements, and used to map systematically the variation of the electrochemical rate constant of a standard [Ru(NH₃)₆]^{2+/3+} probe redox couple with the hBN thickness and dopant level of the composite electrodes using Scanning electrochemical cell microscopy and cyclic voltammetry. The analysis is supported by "Finite element COMSOL Multiphysics simulations". The overarching outcome is the conclusion that nuclear reorganization not only on the solution side of the electrochemical interface but also on the electrode side contributes significantly to the activation (free) energy and rate constant of the electrochemical process. Overpotential dependent electronic contributions to electrochemical interfacial structures, electrocatalysis, and electrochemical processes have been addressed before. The results presented here could, however be of interest, but unfortunately the presentation and whole manuscript are fraught both with some general ("major") and some specific ("minor") shortcomings in the way of stringency and clarity, and in other ways:

Response: We thank the reviewer for their thorough evaluation and thoughtful suggestions.

Some general observations:

- "Defect-mediated charge transfer" makes good sense, and it is understandable that electron or hole transfer between the redox couple and localized electronic states in the electrode material can contribute to the reorganization energy. However, there is no allusion to the precise physical nature of the reorganization energy contribution from the electrode side, the identity of the nuclear modes reorganized, and the nature of the electronic-vibrational coupling. Neither is it clear, how the reorganization (free) energy was calculated, except for frugal allusions to notions such as "continuum model", "simulations"... I am afraid that this is overall very little satisfactory.

Response: We have taken the Reviewer's comments in regard to the clarity of our commu-

nication to heart, and we have implemented changes where we considered appropriate. In the spirit of writing a succinct paper, the details of the theoretical treatment are presented at much greater length in the Methods section of the paper.

Although we believe this is all conveyed in our manuscript, we now address the Reviewer’s specific questions. Our model relies on a dielectric-continuum description, where the reorganization energy of outer sphere electron transfer to a perfect conductor (characterized by a Thomas-Fermi screening length $\ell_{\text{TF}} = 0$) has the following form:

$$\lambda = -\frac{\delta q^2}{4z_0} \left(\frac{1}{\epsilon_\infty} - \frac{1}{\epsilon_s} \right) + \lambda_B \quad (23)$$

The ‘bulk’ term λ_B is the value of reorganization energy away from the electrode. The term $(1/\epsilon_\infty - 1/\epsilon_s)$ is the Pekar factor, and it is a measure of the free energy difference between a charge exclusively solvated by the medium’s fast (electronic) degrees of freedom, and that of a charge fully stabilized by the polarization field’s nuclear and electronic degrees of freedom.

In our treatment, we consider electrodes with finite ℓ_{TF} , and connect this metallicity to the density of states of the material at the Fermi level. At the dielectric continuum level of description, this can be achieved by using the solution to the electrostatic boundary-value problem that arises from considering an electrochemical interface with a non-local (Thomas-Fermi) dielectric response function in the electrode. This analysis results in a modified expression for the reorganization energy:

$$\lambda = \frac{\delta q^2}{4z_0} \left(\frac{\xi_{\ell_{\text{TF}}}(z_0, \epsilon_\infty)}{\epsilon_\infty} - \frac{\xi_{\ell_{\text{TF}}}(z_0, \epsilon_s)}{\epsilon_s} \right) + \lambda_B \quad (24)$$

with a generalized version of the Pekar factor that depends on an image-charge scaling function, ξ . The behavior of the reorganization energy as a function of screening predicted by this theory, and the behavior of the image charges generated in the electrode as a function of metallicity, have also been corroborated by constant potential molecular dynamics simulations³⁰.

These modifications to the reorganization energy are *electronic* in origin. The electrode’s ability to polarize in response to charge transfer renormalizes the electrostatic potential felt by the redox species and hence the statistics of the vertical energy gap, resulting in changes to the reorganization energy of the solvent’s nuclear modes.

• *The reference list to concepts and formalism of fundamental electron transfer theory is misleading. Most of the references listed in the first part of the manuscript are old with little allusion to the developments of electron transfer theory over the past 20+ years or longer. Neither is there any reference to the pioneering work by the Soviet and Russian school, Levich, Dogonadze, Kuznetsov and later by their associates. They could at least refer to the comprehensive review papers and books written by these authors in Western literature. Reference to Chidsey in this context is also misleading, as he simply used formalism developed by the Soviet school.*

Response: We appreciate the reviewer’s observation regarding the foundational electron transfer literature. We recognize that our original citations did not fully capture the seminal

contributions from the Soviet and Russian schools, including the work of Levich, Dogonadze, and Kuznetsov.

We have added the following text to our main text introduction to highlight their seminal contributions: The quantum mechanical theory of ET, pioneered by Levich, Dogonadze, Chizmadzhev, Christov, and Kuznetsov^{26,27}, similarly leads to Marcus–Hush-type rate expressions.

We have also updated our manuscript to include following articles from these foundational works, providing a more balanced and historically accurate context for the theory discussed:

1. Dzhavakhidze, P. G., Kornyshev, A. A. & Krishtalik, L. I. Activation energy of electrode reactions: the non-local effects. *J. Electroanal. Chem. Interfacial Electrochem.* **228**, 329–346 (1987).
2. Phelps, D. K., Kornyshev, A. A. & Weaver, M. J. Nonlocal electrostatic effects on electron-transfer activation energies: some consequences for and comparisons with electrochemical and homogeneous-phase kinetics. *J. Phys. Chem.* **94**, 1753–1761 (1990).
3. Levich, V. G. & Dogonadze, R. R. *Osnovnie voprosi sovremenoj teoreticheskoj elektrokhemii*. In *Proc. 14th Mtg. Int. Comm. Electrochem. Thermodyn. Kinet.* 21 (CITCE, Moscow, 1963).
4. Dogonadze, R. R., Kuznetsov, A. M. & Chernenko, A. A. Theory of homogeneous and heterogeneous electronic processes in liquids. *Russ. Chem. Rev.* **34**, 759–775 (1965).

• *Introduction of background and objectives of the study are only very frugally described, on p.4 lower part.*

Response: Thank you for the valuable feedback. We would like to highlight that detailed motivation and objectives are provided on pages 2 and 3 of the manuscript. While we intentionally omitted some background information to maintain conciseness in accordance with Nature’s guidelines, we have included pertinent and well-chosen citations to situate our study. We are confident that the introduction adequately addresses all essential points relevant to this work.

Our study specifically aims to elucidate the role of the activation free energy in electron transfer (ET) reactions, governed by the key parameter known as the reorganization energy. Traditionally, in heterogeneous ET at electrified solid–liquid interfaces, the reorganization energy has been attributed solely to factors within the electrolyte phase. Here, we demonstrate that the electronic density of states (DOS) of the electrode itself also significantly contributes to the reorganization energy.

Utilizing van der Waals assembly of two-dimensional crystals, we systematically tune the DOS of graphene and measure the resulting effects on outer-sphere ET kinetics. Our results show that the observed variations in ET rates stem from changes in reorganization energy linked to image potential localization within the electrode, which depends on the DOS. This

work establishes the fundamental importance of the electrode’s electronic structure in determining interfacial charge transfer processes. We have elaborated on the experimental design, materials, and analytical methods in detail within the Methods section.

• *I am aware that the theoretical model(s) used may be described in some detail in the Supplementary Material, but a brief overview should also be given in the main text.*

Response: Thank you very much for your helpful suggestion. Given the word limit constraints, we have endeavored to keep our explanations concise while retaining the essential scientific content. In response, we have revised our description to improve clarity for our readers, as highlighted in red below:

“Next, in Figure 3c, we account for this ℓ_{TF} to model a reorganization energy, $\lambda(\mu)$, that is a function of DOS and μ using non-local dielectric continuum theory. **Our model predicts a strong attenuation of $\lambda(\mu)$ with increasing metallicity (increasing DOS), converging to a value of 0.82 eV, consistent with λ_{metal}^1 . This trend arises from enhanced stabilization of the charge transfer transition state due to sharply localized polarization within the electrode.** This manifests in the calculated free energy surfaces for the $\text{Ru}^{3+}/\text{Ru}^{2+}$ redox system at different values of μ (left inset of Figure 3c). These calculations reveal a pronounced influence of electrode screening on electron transfer kinetics: reduced μ monotonically increases the activation free energy (ΔG^\ddagger), corresponding to the intersection of the reactant and product free energy surfaces. The right inset of Figure 3c plots ℓ_{TF} vs. μ . The plot reveals a rapid decline in ℓ_{TF} as the system is doped away from the CNP ($\mu = 0$), directly linking doping-induced Fermi level shifts to screening efficiency. **By explicitly including $\lambda(\mu)$ —which decreases with rising DOS—our model achieves quantitative agreement with experimental rates (red line in Figure 3a), establishing that electrode electronic structure governs λ and by extension the activation free energy landscape. Therefore, the predominant contributor to increasing ET rates with increasing DOS is the decrease in λ , not the change in number of thermally accessible electron donor/acceptor states.”**

Some specific observations:

• *A rationale for the particular choice of multilayer system, graphene, hBN, RuCl_3 . WSe₂ is needed.*

Response: The multilayer system comprising graphene, hBN, RuCl_3 , and WSe_2 was selected for its unique combination of complementary electronic and structural properties. Graphene serves as a high-mobility, conductive platform with tunable carrier density. It also represents a well-defined, simple system where the DOS near the Fermi level can be directly calculated using linear dispersion relations without additional assumptions. hBN acts as an atomically flat, chemically inert spacer enabling precise modulation of doping. RuCl_3 and WSe_2 provide robust charge-transfer doping, which has been extensively documented and widely utilized to modulate graphene’s electronic states without altering its surface morphology.

Following details have been added to **Methods, Section 1.1 Chemicals and materials:**

“We selected the multilayer system comprising graphene, hBN, RuCl_3 , and WSe_2 due to

their complementary characteristics. Graphene offers a tunable and well-defined electronic platform, while hBN serves as an inert spacer that allows precise control of doping. The RuCl₃ and WSe₂ layers function as stable charge-transfer dopants, modulating graphene’s electronic properties without affecting its structural integrity. Together, these materials enable systematic tuning of interfacial doping while preserving the overall structural quality of the heterostructure. ”

• *I wonder if the Butler-Volmer limit is satisfactory, p. 6. The reorganization energy is usually reflected in non-linear effects beyond this limit. This may be inherent in the procedures used, but is not clear and should be stated explicitly.*

Response: We thank the reviewer for the insightful comment. We acknowledge that reorganization energy often manifests as non-linear deviations beyond the Butler–Volmer approximation. However, rigorous quantification of reorganization energy typically requires strategies such as tethering redox species to the electrode to mitigate diffusion effects. Our current work focuses on extracting the heterogeneous k^0 , and evaluating whether the observed enhancements at higher doping levels can be explained solely by an increase in the density of thermally accessible electronic states. Our findings suggest that such enhancements cannot be fully accounted for without invoking modulation of reorganization energy. We agree that future studies directly probing reorganization energy through non-linear effects would provide valuable insights.

• *I notice that the voltammetric shifts in Figure 1c amount to 25-50 mV or thereabout. This is only (1-2) kBT. Is this enough to be significant? And, why is there only a single point in the WSe2 curve in Figure 1d?*

Response: Thank you for the insightful observation. The voltammetric shifts in Figure 1c amount to approximately 100 mV, which corresponds to changes in the k_0 spanning over three orders of magnitude—from about 3×10^{-4} cm/s for monolayer graphene to over 0.5 cm/s for gold electrodes. This variation illustrates how even modest changes in potential can substantially influence electron transfer kinetics in our system.

Regarding the WSe₂ data in Figure 1d, there are two points: one without an hBN spacer and another with a 16 nm hBN spacer. These data demonstrate that modulation of electron transfer kinetics using the work function disparity strategy extends beyond RuCl₃; similar effects can be realized with other solid-state dopants such as WSe₂. The nature and strength of doping depend critically on the magnitude and sign of the work function difference between graphene and the dopant. For example, RuCl₃ acts as a hole dopant with effects persisting up to approximately 70 nm of hBN, whereas WSe₂, an electron dopant, exhibits reduced influence beyond a 16 nm spacer. This highlights how the choice of dopant and its interaction with graphene—as well as the important role of hBN defects in the case of RuCl₃—govern the spatial extent and magnitude of electronic modulation.

• *The discussion on p. 9, both top and bottom is woolly and in need of significantly improved stringency. So is p. 10 top (“...derived from an attenuation.... How was it derived? $\lambda = 0.82$ eV for the redox pair alluded to on p. 9 bottom was not on pure gold but on gold modified by a self-assembled thiol molecular monolayer.*

Response: Thank you for this insightful comment. In response, we have revised our description to improve clarity for our readers, as highlighted in red below:

“Next, in Figure 3c, we account for this ℓ_{TF} to model a reorganization energy, $\lambda(\mu)$, that is a function of DOS and μ using non-local dielectric continuum theory. **Our model predicts a strong attenuation of $\lambda(\mu)$ with increasing metallicity (increasing DOS), converging to a value of 0.82 eV, consistent with λ_{metal}^1 . This trend arises from enhanced stabilization of the charge transfer transition state due to sharply localized polarization within the electrode.** This manifests in the calculated free energy surfaces for the $\text{Ru}^{3+}/\text{Ru}^{2+}$ redox system at different values of μ (left inset of Figure 3c). These calculations reveal a pronounced influence of electrode screening on electron transfer kinetics: reduced μ monotonically increases the activation free energy (ΔG^\ddagger), corresponding to the intersection of the reactant and product free energy surfaces. The right inset of Figure 3c plots ℓ_{TF} vs. μ . The plot reveals a rapid decline in ℓ_{TF} as the system is doped away from the CNP ($\mu = 0$), directly linking doping-induced Fermi level shifts to screening efficiency. **By explicitly including $\lambda(\mu)$ —which decreases with rising DOS—our model achieves quantitative agreement with experimental rates (red line in Figure 3a), establishing that electrode electronic structure governs λ and by extension the activation free energy landscape. Therefore, the predominant contributor to increasing ET rates with increasing DOS is the decrease in λ , not the change in number of thermally accessible electron donor/acceptor states.”**

We also appreciate the correction regarding the specific experimental details of the alluded study. We have modified the text to reflect this subtlety. This system still serves as a reference for the reorganization energy of the $[\text{Ru}(\text{NH}_3)_6]^{+3/+2}$ redox couple at a metallic substrate, which is what it was used for. Updated text:

“First, we consider an MHC model (red dotted line in Figure 3a) in which the reorganization energy (λ_{metal}) is assumed to be a constant $\lambda = 0.82$ eV, as measured for this redox pair on metal (**gold modified by self-assembled monolayer thiols**) electrodes¹.”

• *And, why is the difference between the potential surfaces in the inset of Figure 3c so small, if the reorganization free energy contributions from the electrode are widely different?*

Response: Perhaps the scale, units or size of the inset figure are leading to some confusion, but the free energy surfaces reflect exactly the same changes in the reorganization energy that are depicted in the main plot of Fig. 3(c). As a reminder, in the linear response regime, the reorganization energy can be extracted from the free energy surfaces as half the difference in the minima of the two curves:

$$\lambda = \frac{1}{2} (\langle \Delta E \rangle_A - \langle \Delta E \rangle_B) \quad (25)$$

Fig. 3(c) shows a change in λ of ~ 0.5 eV, which is exactly the difference shown between the minima of the free energy surfaces at $\mu = 0$ and $\mu = 0.5$ eV.

• *The authors have summarized their conclusions in the Discussion and Conclusion Section, p. 11. They could have taken the chance to address also here the precise physical origin of the reorganization features on the electrode side of the interface. The summary should clearly be worked over again.*

Response: We thank the reviewer for this important suggestion. The physical origin of the reorganization energy on the electrode side is already addressed in our manuscript. As stated in our Discussion and Conclusions:

First para- “Instead, we establish that the reorganization energy—particularly at low charge carrier densities characteristic of low-dimensional materials and semiconductors (both bulk and nanocrystalline)—is strongly governed by **the electronic structure of the electrode, scaling inversely with the density of states (DOS).**”

Second para- “More generally, this work motivates a broader exploration of how **local electronic polarizability** impacts chemical reactivity (in both biological and artificial systems) through stabilization of charged transition states.”

These statements encapsulate our central finding that the electrode’s electronic structure plays a decisive role in governing the reorganization energy and, consequently, the electron transfer kinetics.

In conclusion this work could be of interest as noted. However, at best massive re-writing, elaboration, clarification, and revision along the lines suggested is needed.

Response: We thank the reviewer for their detailed evaluation and thoughtful comments. While we appreciate the suggestions provided, we respectfully maintain that some of the proposed changes fall outside the scope or focus of our current study. We believe that the manuscript, as revised, provides a clear, rigorous, and comprehensive presentation of our findings.

Where appropriate, we have clarified points raised and improved the presentation for clarity and transparency. We trust that the manuscript in its current form conveys the significance and novelty of our work effectively. We remain open to addressing any further specific concerns that would help improve the manuscript within the study’s scope.

References

1. Smalley, J. F. *et al.* Heterogeneous electron-transfer kinetics for ruthenium and ferrocene redox moieties through alkanethiol monolayers on gold. *Journal of the American Chemical Society* **125**, 2004–2013 (2003).
2. Zhao, L., Li, Y., Yu, M., Peng, Y. & Ran, F. Electrolyte-Wettability Issues and Challenges of Electrode Materials in Electrochemical Energy Storage, Energy Conversion, and Beyond. *Advanced Science* **10**, 2300283 (June 2023).
3. Kumar, M., Tamang, S. K., Dabi, M., Kumar, A. & Jaiswal, A. Effect of surface charge on wettability and electrolyte behavior on graphene surfaces using molecular dynamic simulation. *Scientific Reports* **15**, 17415 (May 2025).
4. Papaderakis, A. A. & Dryfe, R. A. The renaissance of electrowetting. *Current Opinion in Electrochemistry* **38**, 101245 (Apr. 2023).
5. Fang, T., Konar, A., Xing, H. L. & Jena, D. Carrier Statistics and Quantum Capacitance of Graphene Sheets and Ribbons. *Applied Physics Letters* **91**, 092109 (2007).

6. Van den Beld, W. T. E. *et al.* In-situ Raman spectroscopy to elucidate the influence of adsorption in graphene electrochemistry. *Scientific Reports* **7**, 45080 (Mar. 2017).
7. Brownson, D. A. C., Varey, S. A., Hussain, F., Haigh, S. J. & Banks, C. E. Electrochemical properties of CVD grown pristine graphene: monolayer- vs. quasi-graphene. *Nanoscale* **6**, 1607–1621 (2014).
8. Patel, A. N. *et al.* A New View of Electrochemistry at Highly Oriented Pyrolytic Graphite. *Journal of the American Chemical Society* **134**, 20117–20130 (2012).
9. Adanigbo, P. *et al.* Scanning electrochemical probe microscopy investigation of two-dimensional materials. *2D Materials* **11**, 032001 (2024).
10. Liu, D.-Q. *et al.* Adiabatic versus non-adiabatic electron transfer at 2D electrode materials. *Nature Communications* **12**, 7110 (2021).
11. Wang, Y. *et al.* Modulation Doping via a Two-Dimensional Atomic Crystalline Acceptor. *Nano Letters* **20**, 8166–8172 (2020).
12. Rossi, A. *et al.* Direct Visualization of the Charge Transfer in a Graphene/ α -RuCl₃ Heterostructure via Angle-Resolved Photoemission Spectroscopy. *Nano Letters* **23**, 8000–8006 (Aug. 2023).
13. Biswas, S., Li, Y., Winter, S. M., Knolle, J. & Valentí, R. Electronic Properties of α -RuCl₃ in Proximity to Graphene. *Physical Review Letters* **123**, 237201 (Dec. 2019).
14. Rizzo, D. J. *et al.* Charge-Transfer Plasmon Polaritons at Graphene/-RuCl Interfaces. *Nano Letters* **20**, 8438–8445 (Dec. 2020).
15. Rizzo, D. J. *et al.* Nanometer-Scale Lateral p–n Junctions in Graphene/ α -RuCl₃ Heterostructures. *Nano Letters* **22**, 1946–1953 (Mar. 2022).
16. Kim, K. *et al.* Band Alignment in WSe₂–Graphene Heterostructures. *ACS Nano* **9**, 4527–4532 (Apr. 2015).
17. Nessralla, J. *et al.* Modulating the Electrochemical Intercalation of Graphene Interfaces with α -RuCl₃ as a Solid-State Electron Acceptor. *Nano Letters* **23**, 10334–10341 (Nov. 2023).
18. Velický, M. *et al.* Electron Tunneling through Boron Nitride Confirms Marcus–Hush Theory Predictions for Ultramicroelectrodes. *ACS Nano* **14**, 993–1002 (Jan. 2020).
19. Feldberg, S. W. & Sutin, N. Distance dependence of heterogeneous electron transfer through the nonadiabatic and adiabatic regimes. *Chemical Physics* **324**, 216–225 (2006).
20. Yu, Y. *et al.* Tunable angle-dependent electrochemistry at twisted bilayer graphene with moiré flat bands. *Nature Chemistry* **14**, 267–273 (Feb. 2022).
21. Zhang, K. *et al.* Anomalous interfacial electron transfer kinetics in twisted trilayer graphene caused by layer-specific localization. *ACS Central Science* **9**, 1119–1128 (2023).
22. Inozemtseva, A. I. *et al.* Graphene electrochemistry: ‘Adiabaticity’ of electron transfer. *Electrochimica Acta* **427**, 140901 (2022).

23. Vijaikanth, V., Li, G. & Swaddle, T. W. Kinetics of Reduction of Aqueous Hexaammineruthenium(III) Ion at Pt and Au Microelectrodes: Electrolyte, Temperature, and Pressure Effects. *Inorganic Chemistry* **52**, 2757–2768 (Mar. 2013).
24. Schmickler, W. A theory of adiabatic electron-transfer reactions. *Journal of electroanalytical chemistry and interfacial electrochemistry* **204**, 31–43 (1986).
25. Schmickler, W., Santos, E., Bronshtein, M. & Nazmutdinov, R. Adiabatic Electron-Transfer Reactions on Semiconducting Electrodes. *ChemPhysChem* **18**, 111–116 (2017).
26. Levich, V. G. & Dogonadze, R. R. *Osnovnie voprosi sovremenoi teoreticheskoi elektrokhimii* in *Proceedings of the 14th Meeting of the International Committee on Electrochemical Thermodynamics and Kinetics (CITCE)* (Moscow, 1963), 21.
27. Dogonadze, R. R., Kuznetsov, A. M. & Chernenko, A. A. Theory of homogeneous and heterogeneous electronic processes in liquids. *Russian Chemical Reviews* **34**, 759–775 (1965).
28. Ma, H., Liu, N. & Huang, J.-D. A DFT Study on the Electronic Structures and Conducting Properties of Rubrene and its Derivatives in Organic Field-Effect Transistors. *Scientific Reports* **7**, 331 (2017).
29. Kera, S. & Ueno, N. Photoelectron spectroscopy on the charge reorganization energy and small polaron binding energy of molecular film. *Journal of Electron Spectroscopy and Related Phenomena* **204**, 2–11 (2015).
30. Coello Escalante, L. & Limmer, D. T. Microscopic origin of twist-dependent electron transfer rate in bilayer graphene. *Nano Letters* **24**, 14868–14874 (2024).

Referee report on the manuscript "Electronic origin of the reorganization energy in interfacial electron transfer" by Maroo et al on June 28, 2025.

By using state-of-the-art electrochemical experiments, the authors show notable changes in the outer-sphere electron transfer (OS-ET) kinetics as the substrate underlying a graphene sheet is modulated. In particular, it is found that changing the thickness of a hexagonal boron nitride (hBN) spacer between graphene and RuCl_3 offers a way to control the OS-ET kinetics. An original theoretical model is developed to explain these results in a conceptually novel way and the authors propose that the hBN spacer thickness, i.e. the electrode, controls the OS-ET kinetics by changing the reorganization energy, which is a key parameter in ET theory and which is usually assumed to depend *only* on the electrolyte properties; to my knowledge, this is the first work to propose that rather subtle changes to the *electrode* can strongly influence the reorganization energy and thereby the OS-ET kinetics. This is a notable finding because the conventional wisdom in electrochemistry considers that OS-ET kinetics are usually rather independent of the electrode material (see some references in the comments). While recent studies have challenged this conventional view, especially on graphene electrodes (again, see some references below), variations in the OS-ET kinetics as well as electrode-driven changes in the reorganization energy reported in this work are exceptionally high.

Overall, I consider that this work challenges our current understanding of OS-ET theory and kinetics. Because OS-ET reactions and theories are the corner-stone of all electrochemical reactions, the fundamental understanding gained in the present work may substantially influence both the theory of electrochemical kinetics and the development of novel electrode materials for various electrochemical technologies. The main findings and theoretical model should, however, be more critically compared with previous studies and theories on OS-ET kinetics on graphene-based electrodes. This is needed to place the present work in the broader context of OS-ET reactions on graphene materials and to provide further proof that the reorganization energy is indeed controlled by the electrode potential by eliminating other possibilities.

Major issues

1. The authors present that the observed changes in OS-ET kinetics are due to changes in the reorganization energy resulting from modulations in the electrode electronic structure and the image potential localization in the electrode due to the substrate underlying the graphene surface. Given that the image charge interaction is electrostatic in nature, the observed changes in the electron transfer kinetics appear to be due electrostatic effects. A similar conclusion has been reached for OS-ET on copper-supported graphene electrodes, for which it was shown that changes in the kinetics are also due to variations in the electrostatic potential (<https://www.nature.com/articles/s41467-021-27339-9>). However, this previous work attributes the elec-

trostatic potential changes to changes in the surface charge rather than the electrode metallicity or the reorganization energy. This long prelude leads to a question and a request:

- (a) Further demonstrations of the electrode-controlled reorganization energy is needed. For this it is necessary to consider the option that the changes in OS-ET kinetics could be due to changes in the surface charge rather than the electrode-driven reorganization energy. To resolve whether changes in the electrostatics and OS-ET kinetics are due to changes in the surface charge as proposed previously or the electrode metallicity and the reorganization as proposed herein, I ask the authors to consider additional experiments (e.g. capacitance measurements and/or Kelvin probe force microscopy) to study whether the surface charge varies as a function of the hBN thickness.
 - (b) I ask the authors to carefully compare the findings, methods, and conclusions of the present and previous works (see also references in following points) – I believe this is needed to obtain a deeper perspective and understanding on OS-ET kinetics on 2D electrodes.
2. Related to the previous point, the authors say that the hBN-modulated carrier density shifts the (electro)chemical potential of electrons of the graphene layer. I find this misleading as the electrochemical potential should be to constant across the system at a given electrode potential (on the absolute potential scale, the electrode potential is equal to the Fermi level as well as the electrochemical potential of electrons). Because the electrochemical potential of electrons for a given phase i depends standard part ($\mu^{0,i}$), the electron activity ($a_e^i(r)$), and the electrostatic inner potential $\phi^i(r)$ as $\mu_e = \mu_e^{0,i} - \phi^i(r) + k_B T \ln(a_e^i(r))$, I would rather expect that changing the carrier concentration/density in hBN would influence the electrostatic potential and/or charge distribution within the system. These can e.g. change the filling of the graphene DOS and its surface charge, which would in turn influence the electron transfer kinetics and/or the metallicity of graphene as discussed in point 1. DFT calculations might help to resolve this issue.
 3. In the abstract it is written that "This work challenges a traditional paradigm of heterogeneous ET kinetics and establishes a fundamental role of the electrode electronic structure in interfacial reactivity." I find this misleading: the electrode electronic structure is known to play a key role in interfacial electrochemistry as practically all inner-sphere electron transfer reactions and electrocatalytic reactions depend on the electronic structure. It would be more correct to say that outer-sphere electron transfer kinetics are usually assumed to be independent of the electronic structure (<https://doi.org/10.1016/j.coelec.2019.11.003>, <https://pubs.acs.org/doi/abs/10.1021/acs.chemrev.1c00583>); however, also many

outer-sphere reaction are known to depend on the electronic structure (e.g. <https://doi.org/10.1016/j.coelec.2024.101632>)

4. The authors write that the introduction of RuCl₃ or WSe₂ influence the electrode through the donation of electrons/holes and do not e.g. change the electrode geometry. Is there some proof for this?
5. Figure 2d provides a schematic illustration of the band alignment across the RuCl₃/hBN/graphene systems. In the text this alignment is proposed to result in enhanced ET kinetics due to the downward shift in the Fermi-level and the related increase in the availability of electronic states for ET. The schematic and qualitative picture could be improved and made more quantitative by carrying out DFT simulations of the RuCl₃/hBN/graphene electrode surfaces to study the electrostatic potential and DOS profiles across the system. By further relating the Fermi level with the absolute redox potential of Ru(NH₃)₆^{3+/2+} (see <https://www.nature.com/articles/s41524-023-01184-4>, Figure 5), the authors can strengthen their view on the importance of the higher DOS in increasing OS-ET kinetics.
6. The OS-ET kinetics are modeled using an expression valid for non-adiabatic ET reactions (Eq. 15-16). However, previous work has shown that OS-ET on graphene electrodes for the redox couple considered in this work (Ru(NH₃)₆^{3+/2+}) is adiabatic (<https://www.nature.com/articles/s41467-021-27339-9>). Also, the small DOS at the graphene electrode does not necessarily imply that the ET would be non-adiabatic (<https://doi.org/10.1016/j.electacta.2022.140901>). While the impact of the coupling constant V in Eq.15 will likely cancel out when comparing the ET kinetics across different RuCl₃/hBN/graphene systems and therefore not impact the main conclusion, is there some evidence for assuming that the considered ET reaction is non-adiabatic?

Other issues

- In my opinion, Chidsey's contribution to electron transfer theory cannot be considered seminal as the inclusion of the Fermi-Dirac distribution and the DOS was done well before Chidsey by Levich, Dogonadze, and Kutzentsov et al (e.g. [https://doi.org/10.1016/0079-6816\(75\)90008-8](https://doi.org/10.1016/0079-6816(75)90008-8)). That said, the application of the Marcus-Hush-Chidsey formula by Chidsey in Ref.15 is indeed seminal.
- In general, the reorganization energy is not limited to the solvent nuclei or solvent molecules, and the reorganization energy can depend also on the reorganization of the electrode surface.
- μ is used on page 8 but it is first defined on page 10 as the Fermi level of the system. Furthermore, it would be helpful to be more explicit and call μ as the electrochemical potential of the electrons as commented on in point 2.